# Assessment of Oxidative Stress Markers in Hypothermic Preservation of Transplanted Kidneys

**DOI:** 10.3390/antiox10081263

**Published:** 2021-08-08

**Authors:** Karol Tejchman, Anita Sierocka, Katarzyna Kotfis, Maciej Kotowski, Barbara Dolegowska, Marek Ostrowski, Jerzy Sienko

**Affiliations:** 1Department of General and Transplantation Surgery, Pomeranian Medical University, 70-111 Szczecin, Poland; ktej78@pum.edu.pl (K.T.); maciej.j.kotowski@gmail.com (M.K.); mostrowski@poczta.onet.pl (M.O.); jsien@poczta.onet.pl (J.S.); 2Department of General, Mini Invasive and Gastroenterological Surgery, Pomeranian Medical University, 71-252 Szczecin, Poland; nitus2@wp.pl; 3Department of Anesthesiology, Intensive Therapy and Acute Intoxications, Pomeranian Medical University, 70-111 Szczecin, Poland; 4Department of Microbiology, Immunology and Laboratory Diagnostics, Pomeranian Medical University, 70-111 Szczecin, Poland; barbara.dolegowska@pum.edu.pl

**Keywords:** kidney transplantation, oxidative stress, ischemia-reperfusion injury, enzymes, antioxidants, malondialdehyde, hypothermic machine perfusion, static cold storage, LifePort, outcome

## Abstract

Ischemia-reperfusion injury (IRI) after renal transplantation is a complex biochemical process. The first component is an ischemic phase during kidney storage. The second is reperfusion, the main source of oxidative stress. This study aimed to analyze the activity of enzymes and concentrations of non-enzymatic compounds involved in the antioxidant defense mechanisms: glutathione (GSH), glutathione peroxidase (GPX), catalase (CAT), superoxide dismutase (SOD), glutathione reductase (GR), glutathione transferase (GST), thiobarbituric acid reactive substances (TBARS), malondialdehyde (MDA), measured in preservation fluid before transplantation of human kidneys (KTx) grafted from brain dead donors. The study group (*N* = 66) was divided according to the method of kidney storage: Group 1—hypothermic machine perfusion (HMP) in LifePort perfusion pump, *n*1 = 26, and Group 2—static cold storage (SCS), *n*2 = 40. The measurements of kidney function parameters, blood count, and adverse events were performed at constant time points during 7-day hospitalization and 3-month follow-up. Kidney perfusate in Group 2 was characterized by significantly more acidic pH (*p* < 0.0001), higher activity of GPX [U/mgHb] (*p* < 0.05) and higher concentration of MDA [μmol/L] (*p* < 0.05). There was a statistically significant improvement of kidney function and specific blood count alterations concerning storage method in repeated measures. There were aggregations of significant correlations (*p* < 0.05) between kidney function parameters after KTx and oxidative stress markers: diuresis & CAT, Na^+^ & CAT, K^+^ & GPX, urea & GR. There were aggregations of significant correlations (*p* < 0.05) between recipient blood count and oxidative stress markers: CAT & MON, SOD & WBC, SOD & MON. Study groups demonstrated differences concerning the method of kidney storage. A significant role of recipient’s gender, gender matching, preservation solution, and perfusate pH was not confirmed, however, basing on analyzed data, the well-established long-term beneficial impact of HMP on the outcome of transplanted kidneys might partially depend on the intensity of IRI ischemic phase and oxidative stress, reflected by the examined biomarkers.

## 1. Introduction

Kidney replacement therapy is a general term used to describe medical procedures that help to treat end-stage renal disease (ESRD). It is possible to obtain this goal, with certain limitations, by dialysis; however, the only method for obtaining a healthy organ is kidney transplantation (KTx). A growing number of patients on the transplant waiting lists and an insufficient donor pool remain a significant concern in all countries. There are strategies focused on increasing the number of donations, including the promotion of living donations. Despite donation after brainstem death (DBD), which is the primary source of organs for transplantation, there are an increasing number of extended criteria donor (ECD) and donation after circulatory death (DCD) cases. Improvement of transplantation procedures allowed to overcome past restrictions and possible contraindications. KTx outcomes are improving, and there are many factors involved, including gender [1]. The organs with extended criteria are known to be more susceptible to ischemia-reperfusion injury (IRI), which is a major determinant of delayed graft function (DGF) and related complications, including acute rejection (AR) [2,3,4]. AR and IRI are major causes of graft loss and dysfunction in clinical transplantation [3,5]. Advances in IRI studies regarding its molecular mechanisms are associated with different strategies used to reduce IRI’s detrimental effects. Numerous pathways open the field for therapies against certain points of interest—impairment of endothelium relaxation, scavenging of free radicals, or the blockade of neutrophil activation and adhesion [2]. IRI consists of two phases—ischemia when the blood flow is interrupted for preservation time and reperfusion when the blood flow is restored, the leading cause of oxidative stress. Ischemia results in cell energy depletion and oxidative stress with microcirculatory impairment, inflammation, and apoptosis [6,7,8]. Those two phases involve different organ responses, but the total damage is additive. One of the modifiers of the ischemic phase is the usage of hypothermic machine perfusion (HMP). It is a well-established approach for decreasing the incidence of DGF and improving late outcomes, especially for DCD and ECD, with the advantage of preventing mitochondrial and tissue damage [9,10]. However, perfusion pumps are also widely used for standard criteria donors (SCD). Expected cold ischemic time (CIT) is unpredictable at the moment of procurement due to allocation procedures, and SCD kidney transplantations are known to gain similar advantages from this method of preservation in reducing DGF [11] or AR [12]. On the other hand, in static cold storage (SCS), a massive accumulation of metabolites derived from anaerobic respiration during the ischemic phase increases the hazard caused by oxidative stress in reperfusion [13,14].

Oxidative stress and reactive oxygen species (ROS) generation in the kidney disrupts the excretory function of each section of the nephron. As a result, it makes it impossible for the kidney to compensate for water-electrolyte and acid-base disturbances. In addition, renal regulatory mechanisms are affected: tubular glomerular feedback, myogenic reflex in the supplying arteriole, and the renin-angiotensin-aldosterone system [15]. Oxidative stress is also a broadly emphasized factor in the etiology of many diseases [16,17,18,19] and numerous antioxidative therapies, including the latest COVID-19 related pathologies [20,21].

We hypothesized that there might be a link between oxidative stress and postoperative kidney function parameters and blood count with the method of kidney storage. To prove this hypothesis, we have selected specific biomarkers of established role in oxidative stress: antioxidative enzymes, antioxidants, and markers of lipid damage. Therefore, this study aimed to compare two groups of renal transplant recipients concerning the method of kidney storage: Group 1—hypothermic machine perfusion (HMP) in LifePort, Group 2—static cold storage (SCS), and to oxidative stress biomarkers measured at the end of kidney storage in a cellular material obtained from preservation fluid. It was an attempt to assess the ischemic part of IRI. The study design was based on the determination of three intervals separated in place and time: 1. Preoperative evaluation of donor and recipient-related parameters and their influence on following study intervals; 2. Perioperative evaluation of oxidative stress biomarkers. The general analytic question was if the storage method could be related to the oxidative stress in the ischemic part of IRI; 3. Postoperative evaluation of kidney function and blood count. The first analytic question was whether postoperative parameters are dependent on the method of storage. The second—if they are related to oxidative stress markers. In general, variables from all three intervals were correlated.

## 2. Materials and Methods

### 2.1. Study Group and Inclusion/Exclusion Criteria

The study was conducted within the following time points: A. Donor qualification (DQ), kidney procurement and groups assignment based on the method of kidney storage: Group 1—hypothermic machine perfusion (HMP) and Group 2—static cold storage (SCS), B. Kidney preservation and obtaining kidney perfusate with cellular material at the end of cold ischemic time (CIT) for measurement of oxidative stress biomarkers, C. Recipients qualification (RQ) and kidney transplantation (KTx), D. Hospitalization (H) and 7-day data record including laboratory results for the evaluation of kidney function and blood count, E. Follow-up and laboratory results at 1 and 2 months after KTx. The study flowchart is presented in Figure 1.

In total, 66 donors were qualified for participation in this study after the standard procedure for clinical assessment of brain death (DBD) according to standard criteria (SCD). We did not include DCD donors because there were no such donors in our center when conducting the study. Potentially we could include several ECDs, but their percentage was too low to constitute a comparable group vs. SCDs. Moreover, the vast majority of ECD kidneys in our center were stored in a perfusion pump. Both factors could alter the results in an undesired way. Therefore, ECDs were excluded from the study. Kidney-only and multiorgan donations were included in the study. The standard surgical procedure included the following steps: laparotomy with thoracotomy, aorta and vena cava inferior cannulation, aorta ligation in the abdominal segment, intracorporeal organ flushing with 4 °C preservation fluid: kidney only donation—Custodiol^®^ HTK (Essential Pharmaceuticals, LLC, 1009 Slater Road 210B, Durham, NC 27703, USA), multiorgan donation—CoStorSol™ (Preservation Solutions, Inc., Elkhorn, WI 53121, USA) or SPS-1^®^ (Organ Recovery Systems, Itasca, IL 60143, USA), nephrectomy on both sides with proper tissue margin for storage and transplantation purposes.

The composition of Custodiol^®^ HTK was: 15.0 mmol/L sodium chloride, 9.0 mmol/L potassium chloride, 1.0 mmol/L potassium hydrogen 2-ketoglutarate, 4.0 mmol/L magnesium chloride hexahydrate, 18.0 mmol/L L-histidine hydrochloride monohydrate, 180.0 mmol/L histidine, 2.0 mmol/L tryptophan, 30.0 mmol/L mannitol, 0.015 mmol/L calcium chloride dihydrate, sterile water for injection, anion: 50 mEq Cl^-^. The physical properties of the fluid are as follows: pH 7.02–7.20 at 25 °C (77 °F) [pH 7.4–7.45 at 4 °C (39.2 °F)]; osmolality: 310 mOsm/kg. The composition of CoStorSol™ (University of Wisconsin—UW) was: pentafraction 50 g/L, lactobionic acid (as lactone) 35.83 g/L, potassium phosphate monobasic 3.4 g/L, magnesium sulfate heptahydrate 1.23 g/L, raffinose pentahydrate 17.83 g/L, adenosine 1.34 g/L, allopurinol 0.136 g/L, total glutathione 0.922 g/L, potassium hydroxide 5.61 g/L, sodium hydroxide/hydrochloric acid adjusted to pH 7.4, water for injection, osmolarity 320 mOsm, sodium concentration 29 mEq/L, potassium concentration 125 mEq/L, pH approximately 7.4 at 20 °C. The composition of SPS-1^®^ (UW) was hydroxyethyl starch (HES) 50 g/L, lactobionic acid (as lactone) 35.83 g/L, potassium phosphate monobasic 3.4 g/L, magnesium sulfate heptahydrate 1.23 g/L, raffinose pentahydrate 17.83 g/L, adenosine 1.34 g/L, allopurinol 0.136 g/L, glutathione (reduced form) 0.922 g/L, potassium hydroxide 5.61 g/L, sodium hydroxide 1.16 g/L, potassium hydroxide/hydrochloric acid adjusted to pH 7.4, sterile water for injection (SWI) to 1000 mL. The solution has an osmolality of 310 ± 15 mOsm/kg, sodium concentration of 29 mEq/L, potassium 125 mEq/L, and pH of 7.4 ± 0.1 at 20 °C.

Depending on the type of kidney procurement procedure (one kidney/pair of kidneys/multiorgan) and perfusion pump availability, kidneys were inserted into HMP pump (LifePort^®^, Organ Recovery Systems; Itasca, IL, USA) and assigned to Group 1 (*n*1 = 26), or they were inserted into the standardized container for SCS (large kidney container, product code: H-112, Medans Oy Pihatörmä 1 A, 3. Krs, 02240 ESPOO, FINLAND) and assigned to Group 2 (*n*2 = 40). When conducting the study, the perfusion pump was introduced as the new standard for preservation procedure. Before pumps, all kidneys were preserved in simple cold storage. In the beginning, there was one LifePort available and finally two of them. We took advantage of the situation that there were both types of storage methods available for analysis without forced randomization (the randomization factor was the perfusion pump availability). Kidneys were placed in LifePort or SCS containers immediately after procurement. The perfusion pump had its preservation fluid, specifically designed for this purpose, according to standards. In our study, the standard was PumpProtect^®^ 1000 mL (Carnamedica, POLAND), independent from the perfusion fluid used during organ procurement. It was considered as a potential analytic factor and further discussed. The composition of PumpProtect^®^ was: calcium chloride (dihydrate) 0.5 mmol/L, 4-(2-Hydroxyethyl)-1-piperazineethanesulfonic acid (HEPES, free acid) 10 mmol/L, potassium phosphate (monobasic) 25 mmol/L, mannitol (USP) 30 mmol/L, glucose (anhydrous) beta D (+) 10 mmol/L, sodium gluconate 80 mmol/L, magnesium gluconate (anhydrous) 5 mmol/L, ribose D(–) 5 mmol/L, pentafraction of hydroxyethyl starch (PF) 50 g/L, glutathione (reduced form) 3 mmol/L, adenine (free base) 5 mmol/L, sodium hydroxide qs pH: 7.4 (increasing overall sodium concentration to 100 mmol/L), osmolality 300 mOsm/kg. According to the manufacturer’s manual, systolic pump pressure was set to 30 mmHg and diastolic to 15 mmHg as default values. If the flow could not reach 100 mL/min, the pressure was raised to 35 mmHg and then to a maximal value of 40 mmHg to reach 100 mL/min. The mean flow was 213.4 ± 46.0 mL/min (49–232 mL/min). All kidneys had flow stabilized over 200 mL/min after 3 hours without changing the pressure. Mean resistance was 0.13 ± 0.08 mmHg/mL/min (0.2–0.7 mmHg/mL/min), mean temperature 3.1 °C (2.6–3.5 °C). All parameters were monitored and recorded during the entire preservation period at least once every hour. Kidneys in Group 2 were placed in mentioned containers, filled with preservation solution. The containers were placed in a special transportation box made from thermal insulation material, holding ice at melting temperature, providing 2–4 °C in the chamber. The temperature was maintained, monitored, and recorded during the entire preservation period. LifePort and/or SCS containers were transported to the transplantation department and stored until the surgical procedure.

### 2.2. Samples Collection and Preparation during the KTx Procedure

The procedure of kidney allocation was conducted according to standards governed by the National Transplantation Committee in Poland (Poltransplant). Kidney recipients, earlier qualified by regional centers, were transported and admitted to the nephrology department of a university hospital transplant center and underwent preparation procedures depending on their current medical condition. The following preoperative data were recorded: age, gender, body mass index (BMI), hemodialysis time, panel reactive antibody (PRA), kidney function parameters, blood count.

Kidney perfusate (defined as perfusion fluid after kidney storage) was collected at the surgical department in the operation theatre just before KTx at the end of CIT in both examined groups. The term “perfusate” refers to the preservation solution which was circulating in the kidney (Group 1—HMP) or remained in the kidney during preservation time (Group 2—SCS). In Group 1, it was collected by aspiration from the LifePort chamber just after the kidney was removed for implantation. In Group 2, the perfusate was collected by flushing out the preservation solution through the renal artery. A standard catheter for intravenous infusions was used. Thread head was cut out, and the catheter, connected to an infusion bottle, was inserted into the renal artery. The preservation solution was flushed out by hydrostatic pressure with Ringer’s solution. Perfusate, emerging from the renal vein, was collected into a large 50 mL vial. In samples collection, there was no recipient involvement or violation of the transplant procedure. Flushing out the remaining preservation solution is standard practice before transplantation because its contents are capable of causing potentially severe cardiovascular complications in the recipient, such as hyperkalemic cardiac arrest or bradyarrhythmia. Also, specific components such as allopurinol or pentafraction might cause a hypersensitivity reaction.

Kidney perfusate, independently from the method of collection, always contains remnant morphotic elements, mainly erythrocytes. During the flushing part of organ procurement, blood is gradually diluted and flushed out of blood vessels by a preservation solution, but only to a certain point. The remaining compound is relatively transparent; however, it is cloudy and has a light red/pink color indicating contents of diluted blood. Erythrocytes were crucial for obtaining material for our analysis. We believe that all the cells of procured kidneys underwent oxidative stress; however, erythrocytes were the best material for intracellular measurements of biomarkers activities and concentrations. The perfusate in both groups was preserved with the butylated hydroxytoluene (BHT), which is a lipophilic organic compound widely used to prevent free radical-mediated oxidation in fluids (e.g., fuels, oils) and other materials (final concentration in samples—5 mM). The perfusate was centrifuged at 664 g for 10 min the same way in both groups. There were two sources of perfusate dilution. One was related to the organ flushing mentioned earlier. The second was related to the fact that in static cold storage, perfusate remained in the kidney, while in the perfusion pump, it was flushed to the chamber, and it circulated with a pump solution. Because of this, centrifuged solution in Group 2 contained more morphotic elements (thicker layer). Nonetheless, after obtaining clear, cut separation between fractions, erythrocytes concentration at the bottom of the vial should have been the same in both groups independently from the type of storage and, which was also important, independently from the quality of organ flushing during procurement procedure (in general independently from the dilution before centrifuging). All oxidative stress markers were measured in a homogenous solution containing hemoglobin, obtained from centrifuged cellular material present in the perfusate. We believe there was no need for data normalization to adjust the dilution effect. The fluid over the erythrocytes was removed and frozen at −80 °C for further measurements of perfusate pH. The erythrocytes were washed three times with the PBS solution and frozen at −80 °C in separate vials. Fresh samples were also measured for cross-verification. Results did not differ statistically from those obtained after storage and thawing at room temperature. Double measurements were performed in 30 samples. In both groups, there was enough cellular material. Measurements of stored samples were performed immediately after thawing. The time of thawing did not differ statistically. Basing on the described methodology, we could treat the obtained oxidative stress markers measurements as almost exclusively intracellular (we are aware that a small amount of extracellular perfusion fluid was present in the centrifuged material).

The following KTx procedure was performed according to a standard protocol with the following steps: an approach to retroperitoneal space (preferably on the right side), preparation of external iliac vessels, anastomosis of renal vessels end-to-side, reperfusion, hemostasis, anastomosis of the ureter to the bladder and closure.

### 2.3. Post-Transplant Care

After KTx, the recipients were hospitalized in a surgical transplant department for seven days. During that period, patients were monitored according to standard postoperative procedures. Parameters of kidney function were measured every day: diuresis (24-hour urine collection), plasma concentrations of creatinine, urea, potassium, and sodium. The postoperative observation was complemented with adverse event (AE) records: DGF, AR, and infections (CMV—cytomegalovirus, BKV—virus BK, UTI—urinary tract infection, SSI—surgical site infection). DGF was defined as the need to use dialysis in the first postoperative week. Also, the results of blood count were recorded every day: red blood cell count (RBC), white blood cell count (WBC), lymphocytes count (LYM), monocytes count (MON), platelets (PLT). Afterward, the recipients were transferred to the nephrology department at the same hospital for further post-transplant care and preparation for discharge. After discharge from the hospital, the recipients remained in follow-up in the outpatient transplantation health care center. There was a routine medical visit after 1 and 3 months after kidney transplantation. The same laboratory tests were performed during hospitalization to evaluate the condition of the patient and their graft.

### 2.4. Laboratory Analysis

Red blood cells and clear perfusate were thawed at room temperature. Measurements were performed immediately after thawing. Results in thawed samples were verified. The results in fresh samples did not differ statistically compared to those thawed at room temperature. Double measurements were performed in 30 samples. Reagents kits used in the study are presented in Table 1. The evaluation was made following the guidelines provided by the manufacturer. The activity of enzymes and GSH were calculated per 1 g of erythrocytes’ hemoglobin. Hemoglobin levels were assayed using Drabkin’s method. The principle of the method is that hemoglobin is oxidized by potassium ferricyanide into methemoglobin, which is converted into cyanomethemoglobin by potassium cyanide. The intensity of the color formed was proportional to the hemoglobin concentration in the sample, which was read by spectrophotometer or colorimeter measuring at the wavelength of 540 nm. 1 U (μmol/min) was defined as the amount of the enzyme that catalyzes the conversion of one micromole of substrate per minute under the specified conditions of the assay method.

CAT AK utilized the peroxidic function of CAT. The method was based on the enzyme’s reaction with methanol in the presence of an optimal concentration of H_2_O_2_. The formaldehyde produced was measured colorimetrically with 4-amino-3-hydrazino-5-mercapto-1,2,4-triazole (Pulpard) as the chromogen. Purple color absorbance was read at 540 nm using a plate reader. Formaldehyde concentration was calculated from the linear regression of the standard curve substituting corrected absorbance values for each sample. CAT activity was calculated using the equation provided in the manual. One unit was defined as the amount of enzyme that caused the formation of 1.0 nmol of formaldehyde per minute at 25 °C.

GSH AK utilized an optimized enzymatic recycling method, using glutathione reductase to quantify reduced glutathione (GSH). The main reagent was DTNB (5,5′-dithio-bis-2-(nitrobenzoic acid), Ellman’s reagent), which reacted with the sulfhydryl group of GSH, producing a yellow colored 5-thio-2-nitrobenzoic acid (TNB). The rate of TNB production was directly proportional to this recycling reaction, which was directly proportional to the concentration of GSH in the sample. Therefore, measurements of the absorbance of TNB at 405–414 nm provided an accurate estimation of GSH in the sample. The dynamic range was 0–16 μmol GSH. Oxidized glutathione (GSSG) content was possible to measure by an additional procedure and utilization of 2-vinylpiridine; however, the producer did not guarantee accuracy; hence, it was not performed.

GPX AK measured GPx activity indirectly by a coupled reaction with glutathione reductase (GR). Oxidized glutathione (GSSG), produced upon reduction of hydroperoxide by GPX, was recycled to its reduced state by GR and nicotinamide adenine dinucleotide phosphate (NADPH). The oxidation of NADPH to NADP^+^ was accompanied by a decrease in absorbance at 340 nm (A_340_). Under conditions in which the GPX activity was rate-limiting, the rate of decrease in the A_340_ was directly proportional to the GPX activity in the sample. GR AK measured GR activity directly by measuring the rate of NADPH oxidation, the same as in the GPX AK. The decrease in the A_340_ was directly proportional to the GPX activity in the sample.

The GST AK measured the total GST activity (cytosolic and microsomal) by evaluating the conjugation of 1-chloro-2,4-dinitrobenzene (CDNB) with reduced glutathione. The conjugation was accompanied by an increase in absorbance at 340 nm. The rate of increase was directly proportional to the GST activity in the sample. Absorbance was read once every minute using a plate reader to obtain at least five time points. One unit of enzyme conjugated 1.0 nmol of CDNB with reduced glutathione per minute at 25 °C.

The MDA-586 kit was used to assay total MDA concentration. The assay conditions minimized the interference from other lipid peroxidation products, such as 4-hydroxyalkenals (HNE), which do not produce significant color at 586 nm under the conditions of the assay. The method was based on the reaction of a chromogenic reagent, N-methyl-2-phenylindole (R1, NMPI), with MDA at 45 °C. One molecule of MDA reacted with two molecules of NMPI to yield a stable carbocyanine dye with maximum absorption at 586 nm. The reaction was carried out in hydrochloric acid and with the addition of Probucol, an antioxidant, to further minimize the reaction of HNE. In the assay, a calibration curve was prepared using the MDA standard provided. MDA concentration was determined from the absorbance at 586 nm in the MDA-586 assay and the standard curve.

SOD AK utilized tetrazolium salt for the detection of superoxide radicals generated by xanthine oxidase and hypoxanthine. One unit of SOD was defined as the amount of enzyme needed to exhibit 50% dismutation of the superoxide radical. The SOD assay measured all three types of SOD (Cu/Zn, Mn, and FeSOD). The absorbance at 440-460 nm was read using a plate reader. SOD activity was calculated using an equation obtained from linear regression of the standard curve substituting the linearized rate (LR) for each sample. The dynamic range was 0.005–0.050 units/mL SOD.

TBARS AK assayed lipid peroxidation by measurement of thiobarbituric acid reactive substances, including MDA and other compounds. The MDA-TBA adduct formed by the reaction of MDA and thiobarbituric acid (TBA) under high temperature (90–100 °C) and acidic conditions was measured colorimetrically at 530–540 nm. Values were calculated according to the standard curve, plotted as a linear function of absorbance and TBATS concentration. Under the standardized conditions of the assay, the dynamic range was 0–50 μmol MDA equivalents.

### 2.5. Ethical Approval

This study, No. MB-102-170/16, received approval of the Bioethical Committee of the Pomeranian Medical University in Szczecin (No. KB-0012/77/13, dated 10 June 2013).

### 2.6. Statistical Analysis

Statistical analysis was performed on raw data using Statistica 13 software package (TIBCO Software Inc., 3307 Hillview Avenue Palo Alto, CA 94304, USA). There were no modifications such as normalization, standardization, weighting, etc. Statistical tests were chosen according to the input data and the cause-effect analytic questions. The process was supported by the software. Statistical analysis was divided into four parts. Overall, there were 28 continuous and 26 qualitative variables (HLA—common for both donor and recipient) (Table 2).

Initially, descriptive statistics were performed: frequency tables, histograms, box & whisker plots, scatterplots, means, standard deviations (SD), medians, quartiles, ranges. The default test for normality was Shapiro-Wilk W. Kolmogorov–Smirnov & Lilliefors test was also used, as it is sensitive to differences in distribution. Some variables in our analysis had distribution different normal, and their analysis involved nonparametric tests.

The second part was analyzing the relationship between donor/recipient parameters and oxidative stress biomarkers/kidney function/blood count. Continuous variables were analyzed mainly using Spearman correlation and correlation matrices. The analysis was supported by multiple linear regression (MLR), residual, and Q-Q plots (Figure A1, Appendix B). Qualitative variables analysis involved the comparison of two or multiple independent samples based on The Mann-Whitney U test. 2-group Kolmogorov–Smirnov and Wald–Wolfowitz runs were also used to cross-check if the distributions differed in the variance. Multiple independent samples analysis was performed using Kruskall-Wallis ANOVA & Median. Multiple measurements were analyzed with rANOVA.

The third part of the analysis was the assessment of oxidative stress biomarkers. Measurements were characterized by a relatively high dispersion, thus high SD, yet the distribution was normal. A classical *t*-test of independent groups by variables was performed.

The fourth part was the analysis of kidney function (KF) and blood count (BC). The analysis was conducted according to the “repeated measures design”. The same variables were measured in selected time points—a total of ten measurements. The primary statistical test was rANOVA, which allowed the analysis of multiple independent variables in conjunction with multiple measure time points. Friedman test (Friedman’s ANOVA) was also used additionally as a nonparametric equivalent of one-way ANOVA due to resistance to marginal measurements, normality of distribution, and group abundance. Differences among qualitative variables were analyzed using frequency tables, 2 × 2 tables, and X^2^ test.

The correlations of multiple continuous variables were done using correlation matrices, utilizing the Spearman rank test (Figure A2, Appendix B). Numerous results were presented on color heat maps. More intense color spoke for the greater accumulation of statistically significant results and a tendency towards thoughtful relation between correlated parameters. The intensity of colors was calculated basing on the number of significant (*p* < 0.05, 1 point) and borderline (*p* > 0.05 and *p* < 0.1, ½ point) correlations and upgraded with the indication of positive/negative correlation by hot/cold color gradient respectively. All statistical tests used in the study analysis were supported in Statistica 13 software, and they were used according to selected data sets on input. Statistical significance was set at *p* < 0.05.

## 3. Results

### 3.1. Characteristics of Donors and Recipients Qualified for the Study

Descriptive statistics of donors qualified for organ procurement are presented in Table 3. There were no statistically significant differences between the study groups, considering donor characteristics and grafted kidneys.

Descriptive statistics of recipients qualified for KTx are presented in Table 4. Study groups were analyzed, including the recipients’ characteristics, HLA match, kidney function, adverse events, and immunosuppression. There were significantly more female recipients (*p* = 0.02) and more HLA-DR mismatches (*p* = 0.01) in Group 2. All the remaining differences between the two groups were insignificant. It was noticed that all cases of non-functioning grafts (NGF), acute rejections (AR), and deaths were recorded in Group 2.

The causes of chronic kidney disease in the recipients qualified for KTx are presented in Table 5. There were no statistically significant differences between the study groups. The most frequent cause was glomerulonephritis. Other causes included interstitial bacterial inflammation, hemolytic uremic syndrome, hereditary nephritis, and hyperuricemia. Some causes remained unknown.

### 3.2. Perfusate pH and Donor-Related Parameters

The analysis of the perfusate revealed statistically significant lower pH in Group 2 (*p* = 0.00000001) (Figure 2). All samples of the perfusate pH in Group 2 were below physiologic values (pH < 7.35), and only in 10% of them was the pH above 7.00. On the other hand, in Group 1, the pH of 7.14% of the samples was above 7.35, and in 84.62% of the samples, the pH was above 7.00.

pH was correlated with donor’s related constant variables (Table 2): BMI, creatinine, GFR, CIT (Spearman R, *p* > 0.05, confirmed with Gamma, Kendall T). There was a statistically significant correlation between higher pH and lower donor’s BMI in Group 1 (R Spearman, *p* < 0.05) as well as higher donor’s age in Group 2 (correlation matrix—Pearson test, *p* = 0.045) (Figure 3). A similar analysis in the whole examined group was insignificant. BMI correlation was analyzed in nonparametric tests due to the distribution aggregation towards higher pH values and lower BMI. Donor’s age was analyzed in the parametric Pearson test as a typical linear correlation. There was no other statistical significance confirmed. Additional analyses were presented in Section A.1.

### 3.3. Oxidative Stress Markers in the Ischemic Phase of IRI

The ELISA analyses of oxidative stress biomarkers provided results of antioxidative enzymes activity in units [U] calculated per volume in liters [L] or hemoglobin mass [gHb/mgHb] and non-enzymatic markers concentration in [μmol/L]. All the above measurements were done in cellular material from centrifuged perfusate. The results were presented in Figure 4, Figure 5 and Figure 6 in violin plots to visualize medians, quartiles, and data distribution better. Study groups were marked with color, consistent with the study flowchart (Figure 1). Group 2 was characterized by a statistically significant higher concentration of MDA in the perfusate (Figure 4, Mann-Whitney U test, * *p* = 0.0004). CAT, SOD, and TBARS did not differ statistically between study groups, although in Group 1, there were higher upper quartiles in CAT, the higher median in SOD and TBARS.

GST and GSH did not differ statistically between study groups, although in Group 1 it was noticed higher GST median in the enzyme activity per hemoglobin mass (Figure 5).

Group 2 was characterized by a statistically significant higher mean activity of GPX calculated per hemoglobin mass (Kolmogorov–Smirnov two-sample test, * *p* < 0.05). There were no significant differences in GR activity between study groups (Figure 6).

Perfusate pH was analyzed in study groups with oxidative stress markers. There were statistically significant correlations observed between higher perfusate pH and: higher TBARS concentrations in Group 2 (R Spearman, *p* < 0.05, Figure 7A), lower MDA concentrations in the general group (R Spearman, *p* < 0.05, Figure 7B) as well as higher SOD activity in Group 2 (R Spearman, *p* < 0.05, Figure 7C). Perfusate pH in Group 2 and TBARS concentrations had normal distribution contrary to pH in group 1. Q-Q plots were generated to confirm the adequacy of statistical tests (Figure A1, Appendix B).

### 3.4. Kidney Function Repeated Measures and Oxidative Stress Biomarkers

Kidney function parameters were measured 10 times: before kidney transplant [KTx]—day 0, on day 1,2,3,4,5,6,7, at 1 month, and 3 months). The analysis day 0–7 and months 1–3 was done separately as early function and follow-up function. Urea, creatinine, sodium, potassium, diuresis in 7-day observation were presented in Figure 8. Friedman ANOVA & Kendall’s concordance was used to compare multiple dependent samples (variables) within a measurement row. It answered the question, whether was a statistically significant difference between repeated measurements within study groups. For comparison of the differences between study groups, repeated measures ANOVA (rANOVA) was used. It answered the question of whether varying factors (storage method) influenced kidney function.

Group 1 was characterized by a statistically significant decrease in urea concentration (Friedman, *p* = 0.01028). Group 2 was characterized by a statistically significant increase in urea concentration (Friedman, *p* = 0.00287). There was a statistically significant difference in the dynamics of urea concentration changes in repeated measures after KTx between Group 1 and 2 (rANOVA, *p* = 0.000064) (Figure 8A). Both groups were characterized by a statistically significant decrease in creatinine concentration (Friedman, *p* < 0.000001). Although there was a difference of borderline significance in dynamics of creatinine concentration changes in time after KTx between Group 1 and 2 (Figure 8B), the decrease of creatinine concentration in group 1 was more intense (rANOVA, *p* = 0.055003). Both groups were characterized by a statistically significant decrease in sodium concentration (Friedman, Group 1—*p* = 0.01214, Group 2—*p* < 0.000001). Group 2 was characterized by lowering the sodium concentration below the normal range (135–145 mmol/L) since day 2. In contrast, in Group 1, sodium concentration remained in the normal range (Figure 8C). rANOVA allows to calculate “intercept” (*p* = 0.0421), which gives the information, that all sodium concentration means analyzed together in Group 2 are significantly lower in Group 1. However, the analysis of alterations in repeated measures did not differ statistically between study groups (rANOVA, *p* = 0.46596). Both study groups were characterized by a statistically significant decrease in potassium concentration (Friedman, Group 1—*p* = 0.00015, Group 2—*p* < 0.000001). There was no difference in the dynamics of observed alterations between study groups (rANOVA, *p* = 0.42888) (Figure 8D). Both groups were characterized by a statistically significant increase of diuresis (Friedman, Group 1—*p* < 0.000001, Group 2—*p* = 0.00040). There was no difference in dynamics of observed alterations between study groups (rANOVA, *p* = 0.83546) (Figure 8E). The analysis of repeated measurements at months 1 and 3 was also performed. No statistical differences were found.

There was a statistically significant difference in relative urea concentration changes between day 7 and day 1 (Δ7-1) (t-test for independent groups, CI 95%, Figure 8F). A similar analysis was done for other kidney function parameters and other time points: day 1—day 0 (Δ1-0), month 1—day 1 (Δ1m-1), month 3—day 1 (Δ3m-1). Again, no significant differences between examined groups were observed.

Oxidative stress biomarkers and kidney function repeated measures parameters were analyzed using correlation matrices. Example single chart was presented in Figure A2 in Appendix B. Heat map was created according to the methods described earlier (Table 6). In the general group, there were four strong correlations aggregations: 1. higher CAT and higher sodium, higher diuresis; 2. higher SOD and higher diuresis; 3. higher GST and lower potassium; 4. higher GR and higher urea. There were three medium correlations aggregations in the general group: 1. higher CAT and lower urea, lower creatinine; 2. higher GPX and higher creatinine, higher potassium; 3. higher GR and lower potassium. There were four strong (≥4.0 points) aggregations of statistically significant correlations observed: 1. higher diuresis and higher CAT activity exclusively in Group 1; 2. lower sodium (Na^+^) and higher CAT in Group 2; 3. higher potassium (K^+^) and higher GPX activity in Group 2; 4. higher activity of GR and higher concentration of urea (Ur) tended to aggregate mainly in later measurements (1-3 months) in both groups, while in Group 1 also at day 4–6. There were six medium (2.0-4.0 points) aggregations of statistically significant correlations observed: 1. lower creatinine (Cr) and higher SOD in Group 1; 2. lower sodium (Na+) and higher CAT/higher TBARS in Group 1; 3. lower K^+^ and higher CAT/higher SOD in Group 1; 4. higher diuresis and higher SOD in Group 2; 5. higher Cr and higher GPX/higher GR in Group 2; 6. lower K^+^ and higher GST in Group 2. There was similar aggregation noticed in the general group and Group 1 regarding CAT and GR. There were also a couple of weak aggregations represented by pale colors. In most of the mentioned aggregations, we could observe differences between study groups. Some were just of less intensity; others were not present at all. Detailed analysis of earlier (day 0–7)/later (day 7, month 1, month 3) repeated measurements did not reveal any specific pattern in aggregations.

Recipient gender was analyzed in conjunction with repeated measures of kidney function in study groups. Gender and study groups were categorical qualitative variables for rANOVA. The analysis revealed that repeated measures of urea were statistically significant concerning both mentioned grouping factors (rANOVA, *p* = 0.02138). Together with the analysis conducted earlier (Figure 8A), there were three observations regarding urea: 1. There were statistically significant differences in repeated measures of urea., 2. Mentioned differences were statistically significant between study groups., 3. There were statistically significant differences between genders within study groups: urea concentrations were significantly higher among females in Group 1 and significantly higher among males in Group 2 (Figure 9). Similar analyzes were performed for remaining kidney function repeated measures. No statistically significant differences were found creatinine (*p* = 0.89794), sodium (*p* = 0.71454), potassium (*p* = 0.45501) and diuresis (*p* = 0.90232). Gender was also analyzed in conjunction with qualitative kidney function parameters (Table 2): DGF, AR, death, CMV, BKV, SSI, UTI. Analytic methods were frequency tables, 2 × 2 tables, and X^2^ test. There were no statistical differences confirmed.

### 3.5. Blood Count Repeated Measures and Oxidative Stress Biomarkers

Blood count parameters were measured and analyzed the same way as kidney function. Friedman and repeated measures ANOVA tests gave answers to the same analytic questions (Results 3.4). Both study groups were characterized by a statistically significant increase and then a decrease in white blood cell count (WBC) (Friedman, *p* < 0.000001). The vast majority of recipients developed a leukocytosis. Lower SE borders exceeded the value of 10 G/L in measurements performed at day 1 to day 3. There was no difference in dynamics of the above changes between groups (rANOVA, *p* = 0.69581, Figure 10A). Both study groups were characterized by a statistically significant decrease in red blood cell count (RBC) (Friedman, *p* < 0.000001). The ANVOA “intercept” value of *p* = 0.0032 revealed that overall RBC means in Group 2 were significantly lower. However, the dynamics of their alterations in repeated measures in conjunction with the storage method grouping factor revealed no statistical difference (rANOVA, *p* = 0.88080) (Figure 10B). Both groups were characterized by a statistically significant drop and then an increase of lymphocytes count (LYM) (Friedman, *p* < 0.000001). There was no difference in dynamics of the above changes between groups (rANOVA, *p* = 0.60019) (Figure 10C). Both groups were characterized by a statistically significant drop, increase, and decrease of monocytes count (MON) (Friedman, Group 1—*p* < 0.000001, Group 2—*p* = 0.00063). There was no difference in dynamics of the above changes between groups (rANOVA, *p* = 0.63377) (Figure 10D). Both groups were characterized by a statistically significant decrease of platelets count (PLT) Friedman, *p* = 0.00002). There was no difference in dynamics of the above changes between groups (rANOVA, *p* = 0.47307) (Figure 10E). The analysis of repeated measurements at months 1 and 3 was also done. No statistical differences were found. Relative concentration changes (∆) were analyzed separately (t-test for independent groups, CI 95%). LYM analysis is presented in Figure 10F. Analyzed intervals: day1–day0 (∆1-0), day7–day1 (∆7-1), month1–day1 (∆1m-1), month3–day1 (∆3m-1). Independent group comparison between measurement at day 0 and day 1 revealed significant differences: decrease of LYM (*p* < 0.0001), a decrease of MON (*p* < 0.0001), a decrease of PLT (*p* < 0.0001), however the same comparison of deltas between study groups revealed no significance suggesting, that observed decreases were independent of storage method.

Oxidative stress biomarkers and blood count repeated measures were analyzed the same way as kidney function parameters—correlation matrices & heatmap (Results 3.4). In the general group, there were only two medium correlations aggregations: GST & WBC, MDA & LYM; and minor ones: GR & WBC; TBARS, MDA, GST, GPX & MON; SOD & RBC. They were not represented in study groups. There were three strong (≥4.0 points) aggregations of statistically significant correlations observed (Table 7): 1. Higher CAT and higher MON in Group 1, but at the same time higher CAT and lower MON drop on day 1; 2. Higher SOD and higher WBC in Group 2. Most early measurements were significant. Similar aggregation, but less expressed, was present in Group 1 (medium intensity). In general, higher SOD correlated with higher leukocytosis in both study groups; 3. Higher SOD and lower MON in Group 2. Mostly day 3–7 measurements were significant. There were six medium (2.0–4.0 points) aggregations of statistically significant correlations observed: 1. SOD vs. WBC in Group 1, mentioned earlier. In general, higher SOD correlated with lower MON in both study groups.; 2. Higher MDA and higher LYM & MON. Correlations regarded mainly day 1. With higher MDA, there was a lower drop on LYM & MON after KTx; 3. Higher GST and higher MON ([U/mL] in Group 1 and [U/mgHb] in Group 2), at days 2–7.; 4. Higher GST and lower WBC at day 2–5 in Group 2; 5. Higher GST and higher MON at days 2–7 in Group 2. There were also a couple of weak aggregations represented by pale colors, and in most of the mentioned aggregations, we could observe differences between study groups, similarly as in kidney function analysis. Detailed analysis of earlier (day 0–7)/later (day 7, month 1, month 3) repeated measurements did not reveal any specific pattern in aggregations.

## 4. Discussion

The study procedure provided data from three-time intervals: 1. Preoperative—donor and recipient-related parameters, and perfusate pH, 2. Perioperative—oxidative stress markers (samples for measurements were taken after kidney preservation just before surgical transplant procedure), 3. Postoperative—repeated measurements of kidney function and blood count during 7-day hospitalization and 3-month follow-up. Results have demonstrated the differences in multiple parameters concerning the method of kidney storage. Statistical analysis provided data suggesting that oxidative stress markers may depend on parameters preceding KTx and may influence the outcome. We tried to refer to results subsections concerning current knowledge, reference studies, and possible explanations in the discussion. The study aimed to evaluate oxidative stress markers in the ischemic part of IRI in study groups and verify the hypothesis that kidneys in Group 2 (SCS) underwent greater oxidative stress. Basing on the reference studies, it could be explained by the lack of protective mechanisms of perfusion pumps and more intense internal environment changes in static cold storage. Based on the study flowchart, additional analytic questions were: if the method of kidney storage influenced transplantation outcome expressed in kidney function parameters and if it related to oxidative stress markers. We assumed that kidney procurement & preservation might be the source of oxidative stress, which might modify the KTx outcome, and it might be related to the method of kidney storage where the beneficial effect of HMP on transplanted kidneys might rely on limiting the negative impact of IRI and oxidative stress in the period preceding reperfusion.

### 4.1. Perfusate pH (Figures 2 and 3)

Perfusate pH was assessed in a fluid fraction after centrifuging. Differences between study groups were expected phenomena, mainly due to the dilution effect. pH was significantly lower in Group 2 (SCS), where perfusate was obtained by flushing the static preservation solution out of the kidney. During the entire period of cold storage, the solution remains in the graft since the procurement. Metabolites accumulated with time, including hydrogen ions (H^+^) (as lactic acid). Kidney vascular bed volume in SCS was not exchanged with the outside, contrary to the perfusion pump. Despite cooling significantly reduces the rate of anaerobic metabolism, ongoing hypoxia is a source of metabolic acidosis [22]. Lowered pH and an accumulation of toxic products of anaerobic respiration contribute to free radical generation and damage upon reperfusion of the organ with recipient blood [23,24]. It is also well known that the severity of IRI affects DGF incidence after kidney transplantation [25].

We assumed that the level of graft remnant metabolism was similar in both groups since the storage temperature was the same. H^+^ concentration in static cold storage was higher than in a perfusion pump because it was not diluted by pump preservation solution. Centrifuging did not affect the fluid part of perfusate. Acidosis and DGF incidence is reported to be proportional to CIT [26]. Preventing intracellular acidosis is a critical property of an effective organ preservation solution. Buffer capacity and efficiency depend in a large proportion on pH and temperature [27]. Preventing DGF, in general, is essential for improving the outcome after KTx [28,29]. Despite reference studies supporting the relation between acidosis, CIT, and DGF, we could not confirm the correlation in the present study and a more recent trial [30]. Additional analyses and commentary were presented in Section A.1.

### 4.2. Oxidative Stress Markers in the Ischemic Phase of IRI (Figures 4–6)

Oxidative stress markers were crucial to compare oxidative stress levels in the ischemic part of IRI between the study groups and to identify their possible relationship to the method of kidney storage and other study variables. They were assessed in condensed cellular material (mainly erythrocytes) obtained after perfusate centrifuging. This part of the analysis was performed to evaluate the state of oxidative markers in the particular time-point in the ischemic phase of IRI and determine differences between study groups for further correlations. Reactive oxygen species (ROS) and reactive nitrogen species (RNS) are highly implicated in all kinds of kidney injuries, including IRI. Antioxidant defense mechanisms comprise independent regulation of cytoplasmic, mitochondrial, and nuclear ROS levels [31]. In addition, those levels are regulated by numerous antioxidant enzymes, including CAT, SOD, GPX, peroxiredoxin (Prx), thioredoxin (Trx), and cytochrome c oxidase [32,33].

CAT is primarily distributed in the cytoplasm of proximal tubules of the juxtamedullary cortex [34]. Its deficiency has been reported to increase tubulointerstitial fibrosis and the lipid peroxidation products of tubulointerstitial lesions in unilateral ureteral obstruction mice [35]. In addition, it plays a vital role in decreasing oxidative stress by regulating intrarenal renin-angiotensin system (RAS) and peroxisomal metabolism [36]. Therefore, we hypothesized that Group 2 underwent greater oxidative stress in the ischemic part of IRI due to significant metabolites accumulation. Therefore, CAT was expected to have higher activity; however, no difference was observed.

SOD was numerously reported as protective against diverse kidney diseases [37]. Its dysfunction has been reported to aggravate renal dysfunction, tubulointerstitial fibrosis, inflammation, and apoptosis in the kidney [38]. Several research experiments have provided evidence that deleting or overexpressing SODs through genetic manipulation or medication changes oxidative stress and the disease severity of acute kidney injury (AKI) or chronic kidney disease (CKD). All three SOD isoforms play a crucial role in the deterioration and alleviation of various kidney diseases [37]. In our study, SOD was also expected to have higher activity in Group 2 for the same reasons as CAT, however, it was not confirmed.

TBARS and MDA were selected as low-molecular-weight end products formed via lipid peroxidation. We expected to observe higher concentrations of those markers in Group 2 due to anticipated oxidative stress and more significant cellular damage. However, TBARS did not differ statistically between study groups. It may be explained that TBA-reactive material is questioned in modern evaluation because most of it in human body fluids is reported not to be related to lipid peroxidation. MDA concentration, on the contrary, was significantly higher in Group 2. It is neither the sole end-product of fatty peroxide formation and decomposition nor a substance generated exclusively through lipid peroxidation. However, the difference observed could speak for increased oxidative stress in kidneys in Group 2. MDA levels were reported to be considerably higher in patients developing DGF [39]. Elevated levels of MDA and reduced antioxidant activity have been reported in renal patients [40,41,42,43]. MDA levels elevation as an oxidative stress marker seems to be connected with increased inflammatory state probably derived from a pre-activation state of immune cells [44]. It is also a predictor of undergoing age-associated decay, either aging recipient and kidney graft itself [45]. Certain studies have reported increased systemic biomarkers of oxidative stress in KTx recipients, particularly in the early phase [46,47]. Moreover, oxidative stress and lipid peroxidation appear to be firmly grounded in all senile pathological and dysfunctional conditions.

GST was reported to target antioxidant defense mechanisms against oxidative stress, e.g., neurodegenerative diseases [48]. GST is not a kidney-specific or kidney-related enzyme. Under oxidative stress conditions, GST isoform omega-1 overexpression has been found to increase the resistance to oxidative damage [49]. By above, GST activity in our study was expected to be higher in Group 2; however, no significant differences were observed.

In the study analysis, there was a particular interest in GSH since it is primarily responsible for the cellular antioxidant capacity. In healthy cells and tissues, more than 90% of the total glutathione pool is in the reduced form (GSH), with the remainder in the disulfide form (GSSG) [50]. The quantity of GHS represents the cell resources to actively “disarm” a certain amount of ROS during oxidative stress. GSH in our study was expected to be lower in Group 2 due to oxidation. GSH depletion expressed in the GSSG-to-GSH ratio is a measure of increased cellular oxidative stress [51,52]. Nonetheless, GSH revealed no difference between study groups.

Finally, two enzymes responsible for GSSG-GSH turnover, glutathione-disulfide reductase (GR) and glutathione peroxidase (GPX), were analyzed. The cell’s antioxidant capacity is maintained in favor of GSH due to GR, which regenerates GSH from GSSG [53]. It cooperates with GPX to reduce lipid hydroperoxides to their corresponding alcohols, reduce free hydrogen peroxide to water, and protect the organism from oxidative damage [54]. GPX1 isoform predominantly exists in normal kidneys, accounting for 96% of kidney GPX activity, and was postulated to play a principal role in protecting kidneys from oxidative stress [55]. Both enzymes were expected to be elevated in Group 2. GPX met those expectations, while GR activity [U/gHb] revealed no differences. However, higher median and higher upper quartile were noticed. Overall, from all examined oxidative stress markers, only MDA concentration and GR activity were higher in Group 2. It would be too broad to claim that machine perfusion is the primary factor in limiting oxidative stress in the ischemic phase of IRI; however, those results might support the hypothesis that preserved kidneys underwent greater oxidative stress in static cold storage.

### 4.3. Oxidative Stress Markers and Perfusate pH (Figure 7)

Perfusate pH was expected to correlate with oxidative stress markers. We wanted to verify if the intensity of acidosis might express the severity of oxidative stress resulting in increased levels of antioxidative enzymes, increased TBARS and MDA, and decreased GSH. The analysis revealed three significant correlations—more intense acidosis with lower TBARS, lower SOD, and higher MDA. Only MDA concentrations were coherent with the hypothesis. Oxidative stress was measured intracellularly and pH in the extracellular compound, not reflecting the cellular acid-base balance. TBARS and SOD were lower with lower pH. Reports confirm acidosis-induced production of TBARS, and the results suggest that the contribution of acidosis to peroxidative damages is probably less important than other cytotoxic mechanisms [56]. Some studies suggest that acidic pH may actually protect against hypoxic injury [57,58,59]. Thus, it could support the contrary hypothesis that lipid peroxidation and oxidative stress intensity might be lower with more significant acidosis. Overall, we could not firmly support the claim that ischemia and acidosis primarily influenced oxidative stress in the examined time point.

### 4.4. Kidney Function Parameters in Repeated Measurements (Figure 8)

Repeated measures of kidney function parameters revealed significant improvement in the following days after KTx. Kidney after transplantation goes through a phase of homeostasis restoration and adaptation, which results in functional improvement. We observed a gradual decrease in creatinine and potassium and a gradual increase in diuresis in both groups, which accounts for the recovery of excretory capabilities and primary kidney function. Creatinine and diuresis had linear characteristics, while potassium had the phase of increase at day 2. Medications used frequently after kidney transplantation, including calcineurin inhibitors, angiotensin-converting enzyme inhibitors, angiotensin receptor blockers, beta-blockers, and antimicrobials, are considered the leading culprit for posttransplant hyperkalemia in recipients with a well-functioning allograft [60]. Lingering hyperkaliemia may have a cause in high pre-transplant plasma potassium [61] or may be caused by postoperative complications and immunosuppression [62]. Sodium in our measurements had a decrease phase, then stabilization since the measurement at day 2, which related to potassium decrease. Detailed ANOVA analysis revealed no differences between study groups in alterations of all the above parameters. However, it was noticed that in Group 2, in measurement at day 2 and following, sodium was stabilized at an average concentration of about 134 mmol/L (below lower physiologic border). At the same time, in Group 1, there was a trend for an elevation above 135 mmol/L, which might be related to the storage method. KTx recipients have an impaired ability to dilute urine, and hyponatremia is a risk factor for adverse events in CKD and after KTx. Reduced osmoregulation performance occurs frequently and is an independent predictor of renal outcome [63]. We believe that data dispersion and group size did not allow to meet a statistical significance between the study groups in kidney function improvement (creatinine revealed borderline significance, *p* = 0.055), however, we could interpret the results and tendencies in graphic representation as a possible beneficial effect of kidney machine perfusion on kidney function.

In the above analysis, one result was unexpected—the urea. In Group 1, the mean concentrations decreased in the following measurements, while in Group 2, they increased. Studies report that urea serum concentration and urinary excretion are inversely related to graft failure and mortality [64]. Urea is a marker of protein turnover and kidney function. The cycling of and excretion of urea by the kidneys is a vital part of metabolism. Besides its role as a carrier of waste nitrogen, urea also plays a role in the countercurrent exchange system of the nephrons, which allows for the reabsorption of water and critical ions from the excreted urine. Mentioned mechanisms are strictly connected with primary kidney function. Higher or increasing mean urea concentrations in Group 2 might be explained by more significant kidney functional impairment in static cold storage. It remains unclear why urea differences between the study groups are expressed contrary to other kidney function parameters.

### 4.5. Kidney Function Parameters and Oxidative Stress (Table 6)

Oxidative stress markers were correlated with kidney function parameters. It was crucial to answering whether oxidative stress influences the graft function and whether it is related to study groups. The strongest aggregation in matrices correlation was observed between higher CAT and higher diuresis in Group 1 (HMP). Similar aggregation was also noticed in the general group, where additionally higher SOD correlated with higher diuresis. The result was unexpected. We would anticipate lower diuresis with higher oxidative stress; however, it would be hard to draw a reliable conclusion basing on only those two elevated enzymes activities in correlation with higher diuresis. In the general group, there were also strong correlations aggregations between higher GR and higher urea as well as medium ones between higher GPX and higher creatinine/higher potassium. Those could support the hypothesis that higher oxidative stress might cause worse graft function; however, those correlations were not supported by other oxidative stress markers. Moreover, higher GR and GST correlated with lower potassium. Overall, the results were ambiguous.

Similar observations were present in study groups. First, there was a strong aggregation between higher CAT and lower Na^+^. In this case, aggregation was represented in both groups. Moreover, it somehow reflected alterations of Na^+^ in measurements from day 2—higher means in Group 1, lower means in Group 2. Thus, it could support the hypothesis that Group 2 underwent greater oxidative stress than Group 1 due to worse osmotic regulation; however other biomarkers did not support Na^+^ correlations. Next, there was a strong aggregation between higher GPX and lower K^+^ in Group 2, and again—it was not supported by a similar one in Group 1 and from other biomarkers. Last strong aggregation—higher GR and higher urea in both groups, yet with no similar support from other biomarkers. Despite mentioned strong aggregations, there were several more medium and minor ones (Table 6). However, in the vast majority of them, there was a problem of unequivocal interpretation. Details were provided in Section A.3. The results presented on the heatmap could potentially be used to anticipate the outcome basing on the oxidative stress markers; however, none of the aggregations could fulfill conditions allowing to support the hypothesis that kidney function after KTx was dependent on oxidative stress markers measured in the ischemic part of IRI.

### 4.6. Gender and Kidney Function in Study Groups (Figure 9)

Regarding recipient gender and kidney function, rANOVA provided detailed results on how kidney function parameters change in repeated measures in conjunction with two grouping factors (recipient gender and study groups). The analysis did not reveal additional statistically significant differences, as did the one-factor research with study groups. There was only one exception—urea. The analysis revealed that in Group 1, females are characterized by higher concentrations of serum urea. In Group 2, it was quite the opposite. It is hard to say whether its independent impact caused the gender effect or coincidentally caused by a specific, explicit difference in serum urea between study groups. Unfortunately, the mathematical calculation did not allow the extraction of this information on the present study group.

### 4.7. Blood Count in Repeated Measurements (Figure 10)

Blood count analysis in postoperative repeated measures revealed that all parameters (WBC, RBC, LYM, MON, PLT) had statistically significant alterations within both examined groups. However, the analysis of differences between study groups was insignificant. All recipients had a significant increase in WBC count in the first measures after KTx. Most of them developed leukocytosis over 10 G/L, caused by neutrophils basing on the equation NEU ≈ WBC—LYM—MON. Leukocytosis is a natural defense mechanism against general stress applied to the organism. Leukocytosis occurs after physical training [65], labor and delivery [66], elective surgery [67,68] as well as after organ transplantation [69,70]. In most cases in our study, leukocytosis returned to normal on the 4th day after KTx.

In both groups, we observed a significant decrease in RBC. In our transplantation center, average blood loss during KTx usually does not exceed 300 mL (range 100–300 mL), similar to the reported minimally invasive laparoscopic KTx procedure [71]. Postoperative surgical-site hemorrhages are not frequent (4.6%) and usually occur during the first day postoperatively [72]. In our study, there was no such event reported, which would have required surgical intervention. However, there were cases where surgical site catheter was maintained due to blood collection of more than 100 mL per 24 h. Postoperative blood loss usually did not exceed 400 mL, and surgical site catheter was usually kept no longer than 4 days. Therefore, we believe that the RBC decrease in our study group might result from perioperative blood loss and intravenous fluid therapy.

On the 1st day after KTx, we observed a significant drop in LYM and MON in both groups. LYM gradually increased during the following days, while MON increased on the 2nd day and slowly decreased. In our study, more than half of the recipients in both groups received induction therapy in immunosuppressive treatment—Simulect or ATG, agents that profoundly deplete lymphocytes. Such therapy involves up to 83% of renal transplants worldwide, and lymphocyte depletion before or beginning at the time of transplantation is beneficial in reducing maintenance immunosuppression [73]. Since the lymphocyte-depleting antibodies were administered only during the induction of the immunosuppressive therapy, the lymphocytes population was rebuilt with time in both groups. Monocytes are reported to significantly contribute to ischemia-reperfusion injury and allograft rejection after kidney transplantation; however, knowledge about the effects of immunosuppressive drugs on monocyte activation is limited [74]. Studies suggest that blood leaving the kidney graft is rich with CD4^+^ lymphocytes and CXCR3 monocytes, indicating the presence of inflammatory activity [75]. In the lungs, transplantation monocytes and macrophages mediate both the pathogen response and sterile lung inflammation, including that arising from IRI [76].

In both groups, we also observed a significant decrease in PLT. Their deposition in the kidney is related to IRI and AR [77]. The experimental data from the last decade demonstrate that platelets contribute to acute vascular inflammation and atherosclerosis and can modify innate and adaptive immune responses to transplants. New findings demonstrate heterologous interactions of platelet microparticles with leukocytes that may increase their range of impact. By attaching to neutrophils, platelet microparticles migrate out of blood vessels and stimulate cytokine secretion [78].

### 4.8. Blood Count and Oxidative Stress (Table 7)

Correlations between oxidative stress and blood count were determined using heatmaps and correlations matrices. Significances were not as anticipated as correlations of perfusate pH and kidney function since blood count interferes with numerous factors. However, possible relations were analyzed the same way as kidney function parameters and presented in Table 7. The interpretation of correlations between oxidative stress markers and kidney function underwent similar limitations to the kidney function parameters mentioned above. The analysis in study groups revealed some differences. We could point to three aggregations that could get closer to being considered influential: SOD & WBC; SOD, CAT & MON. Observed correlations had the representation in both examined groups; however, they were not expressed in the general group. Basing on the above, it would be hard to conclude that oxidative stress biomarkers alterations were significant independent factors modifying blood count. We are aware that many other factors are influential during the KTx procedure, including major ones: WBC—operative stress, RBC—bleeding, LYM & MON—immunosuppression, PLT—microaggregation.

Overall, we were able to demonstrate only some differences in the general group and between groups concerning oxidative stress markers and some correlations between oxidative stress and kidney function as well as blood count. We might expect more statistical significance since oxidative stress is a crucial part of IRI, which strongly influences kidney function and the outcome. However, we must take into consideration that in our study, we evaluated the ischemic part of IRI. Oxidative stress markers were measured before reoxygenation. We believe this was a critical factor in our analysis. On the one hand, we are aware that it was one of our study limitations, but on the other, results might shed some light on the role of the cold ischemic period in oxidative stress.

### 4.9. Limitations

When designing our study, we took advantage of the situation that we were introducing perfusion machines for kidneys preservation in our transplantation center. Thus, it was the opportunity to compare the examined groups. Before that, all kidneys were kept in SCS. With time two LifePorts were operational, and since then, most of the kidneys were preserved by HMP. In total, 66 recipients were included in the study (*n*1 = 26 in Group 1, HMP; *n*2 = 40 in Group 2 SCS), which was enough to observe reported statistical differences, yet the subgroups were relatively too small to reach all the desired significances from multifactorial analysis and correlation matrices. Research widely supported the selection of biomarkers for our study. Indicators of lipid peroxidation could be extended by, e.g., markers of protein carbonylation. The process is one of the most harmful irreversible oxidative protein modifications. Nonetheless, the assessment requires proteomic quantification of protein-bound carbonyls with metal-catalyzed oxidation, lipid peroxidation, glycation/glycoxidation, or mass spectrometry-based techniques. We chose to examine the ischemic part of IRI, where material for measurements of oxidative stress biomarkers was obtained at the end of kidney storage. Reference research analysis revealed that this part of IRI was not commonly chosen for the study. Nevertheless, we found that acute hypoxia during procurement may be the source of considerable oxidative stress. We are aware that the reperfusion part of IRI and the following oxidative stress substantially exceeds the one from preservation. Therefore, it might provide valuable data for comparing the status before and after reperfusion. However, available procedures were limited at the time by the transplantation protocol, recipients’ consent, and bioethics committee approval. Other time points from the preservation time were also considered, but it would violate standards of kidney storage. From the analytic perspective, it would be valuable information to compare at least two-time points. In the case of HMP, we would have to open the kidney chamber and risk contamination. In SCS, we would have to flush out preservation fluid out of the kidney and stop proper preservation before Tx time. In our case, the method was safe and without the need to violate the protocol. Histological evaluation would have been a valuable addition to our analysis; however, a protocol biopsy is not a standard approach in Poland.

## 5. Conclusions

Kidneys preserved in static cold storage (SCS) suffered significantly more intense acidosis measured in extracellular fluid. Increased acidosis significantly correlated with higher donor BMI and lower donor age. In the SCS group, there was a considerably higher mean concentration of MDA and higher mean activity of GPX, suggesting more intense oxidative stress. There were statistically significant correlations between oxidative stress markers and perfusate pH; however, they did not confirm that oxidative stress is directly related to the observed acidosis. There was a statistically significant improvement of kidney function in repeated measurements, suggesting that hypothermic perfusion pump (HMP) storage might have some advantages concerning the outcome. Study groups were characterized by the statistically significant difference in the dynamics of urea concentrations alteration in the postoperative period.

There were statistically significant correlations between oxidative stress markers and kidney function; however, they could not confirm that kidney function after transplantation is related to oxidative stress markers measured in preservation time. There were certain alterations of blood count observed in repeated measures after KTx; however, they seemed to relate to other factors than oxidative stress or acidosis. Despite few statistically significant relations, the influence of recipient gender, gender matching, preservation solution, and perfusate pH was not confirmed.

## Figures and Tables

**Figure 1 antioxidants-10-01263-f001:**
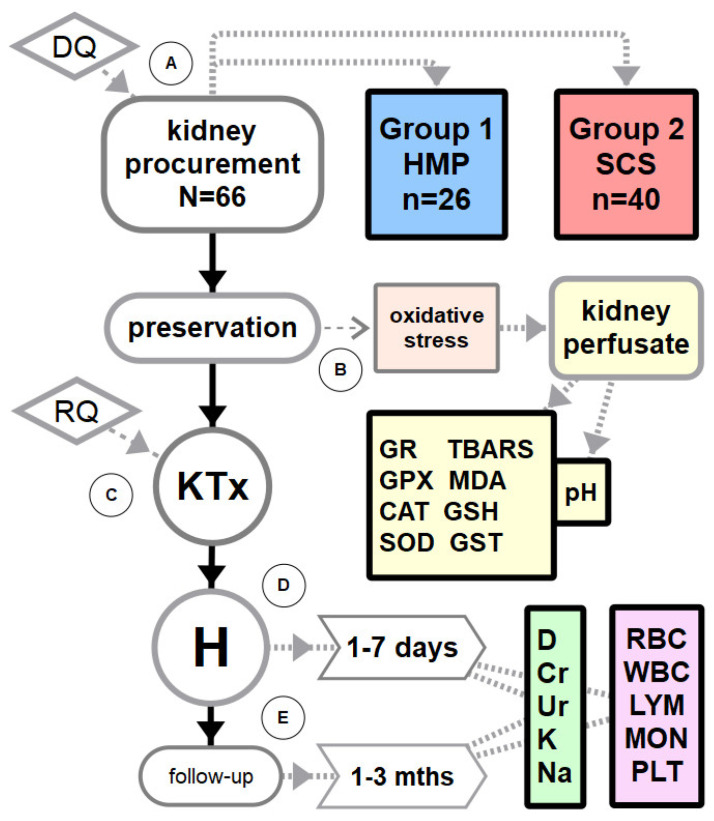
Study flowchart. Legend: DQ—donor qualification, HMP—hypothermic machine perfusion, SCS—static cold storage, RQ—recipient qualification, KTx—kidney transplantation, H—postoperative hospitalization, mths—months, D—diuresis, Cr—creatinine, Ur—urea, K—potassium, Na—sodium, RBC—red blood count, WBC—white blood count, LYM—lymphocytes, MON—monocytes, PLT—platelets.

**Figure 2 antioxidants-10-01263-f002:**
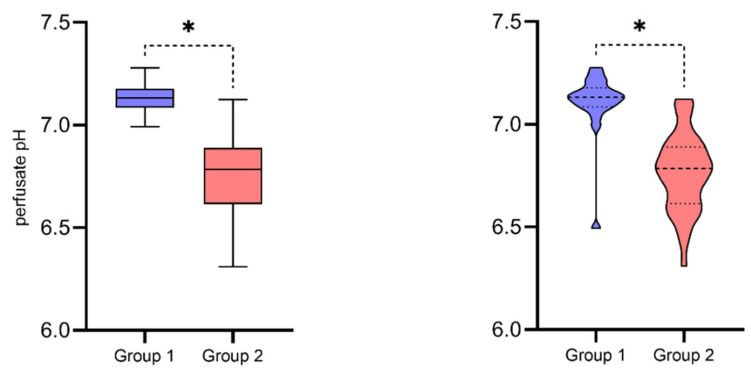
Perfusate pH in the study groups means & standard deviations; violin plots—medians, quartiles, and distribution. (Mann-Whitney U test, Group 1 (HMP) *n*1 = 26 vs. Group 2 (SCS) *n*2 = 40, *N* = 66, *** *p* = 0.00000001**).

**Figure 3 antioxidants-10-01263-f003:**
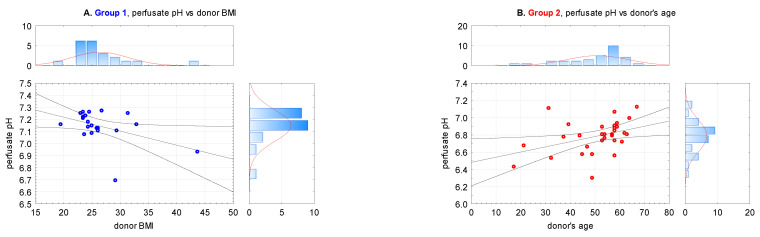
Scatterplots of correlations between perfusate pH and: (**A**). BMI in Group 1 (HMP) (R Spearman, ***p* < 0.05**, *n*1 = 26), (**B**). donor’s age in Group 2 (SCS) (Pearson test, ***p* = 0.045**, *n*2 = 40).

**Figure 4 antioxidants-10-01263-f004:**
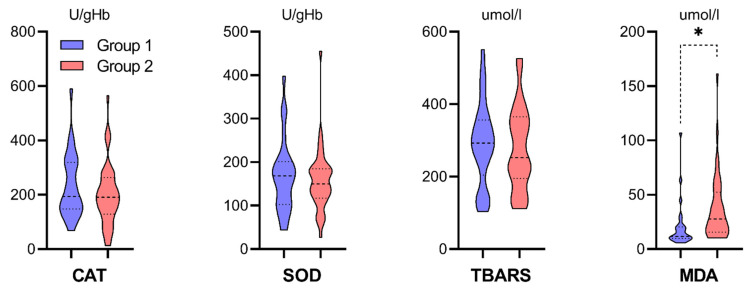
Violin plots of distribution (Mann-Whitney U test, *N* = 66, *n*1 = 26, *n*2 = 40): CAT activity (*p* = 0.3713), SOD activity (*p* = 0.7007), TBARS concentration (*p* = 0.6158), MDA concentration (* ***p* = 0.0004**).

**Figure 5 antioxidants-10-01263-f005:**
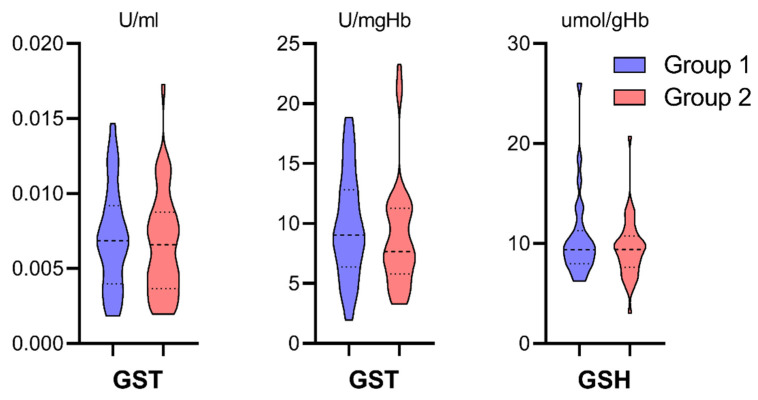
Violin plots of distribution (Mann-Whitney U test, *N* = 66, *n*1 = 26, *n*2 = 40): GST activity per volume (*p* = 0.6250), GST activity per hemoglobin mass (*p* = 0.2639), GSH concentration (*p* = 0.5357).

**Figure 6 antioxidants-10-01263-f006:**
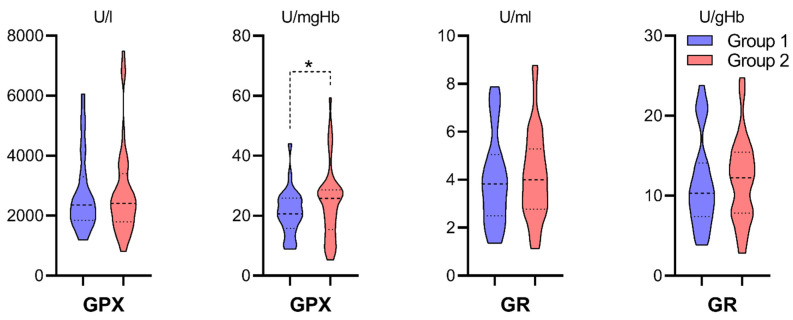
Violin plots of distribution (*N* = 66, *n*1 = 26, *n*2 = 40): GPX activity per volume (Mann-Whitney U test*, p* = 0.9429), GPX activity per hemoglobin mass (Kolmogorov–Smirnov two-sample test, * ***p* < 0.05**), GR activity per volume (Mann-Whitney U test*, p* = 0.5950), GR activity per hemoglobin mass (Mann-Whitney U test, *p* = 0.9969).

**Figure 7 antioxidants-10-01263-f007:**
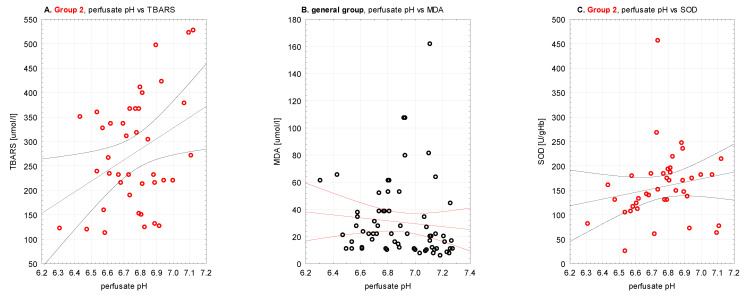
Scatterplots of correlations between perfusate pH and: (**A**). TBARS concentration in Group 2 (SCS) (R Spearman, ***p* < 0.05**, *n*2 = 40), (**B**). MDA concentration in the general group (R Spearman, ***p* < 0.05**, *N* = 66), (**C**). SOD activity in Group 2 (SCS) (R Spearman, ***p* < 0.05**, *n*2 = 40).

**Figure 8 antioxidants-10-01263-f008:**
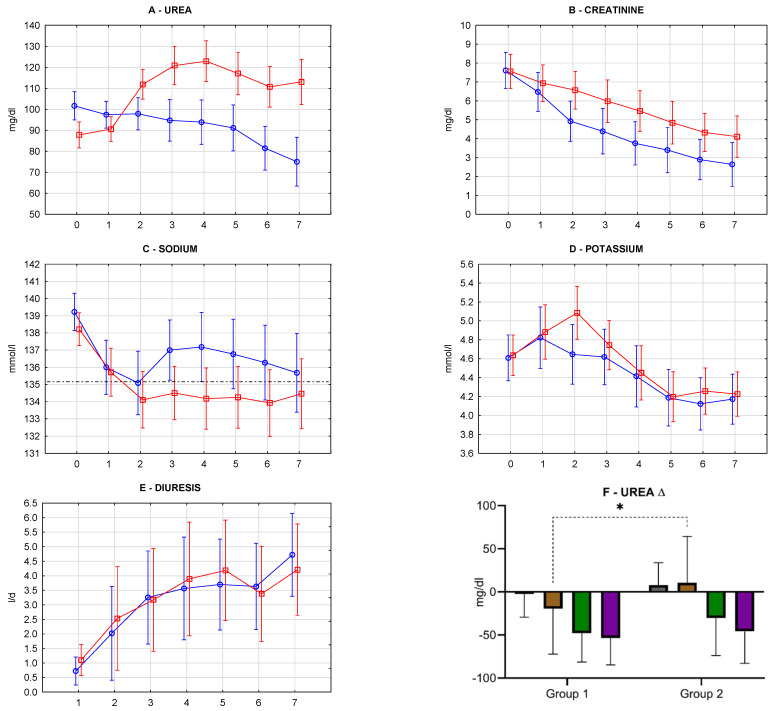
Mean kidney function parameters in following measures—7-day observation (rANOVA, *N* = 66, 
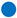
 Group 1 (HMP) *n*1 = 26, 

 Group 2 (SCS) *n*2 = 40), T—standard error (SE)): (**A**). urea (*** *p* = 0.00006**), (**B**). creatinine (*p* = 0.055), (**C**). sodium (*p* = 0.466), (**D**). potassium (*p* = 0.429), (**E**). diuresis (*p* = 0.835). (**F**). urea concentration relative changes (Δ), (*t* test for independent groups, CI 95%): 
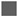
 day1–day0 (*p* = 0.135), 
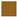
 day7–day1 (*** *p* = 0.029**), 
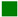
 month1–day1 (*p* = 0.087), 

 month3–day1 (*p* = 0.643).

**Figure 9 antioxidants-10-01263-f009:**
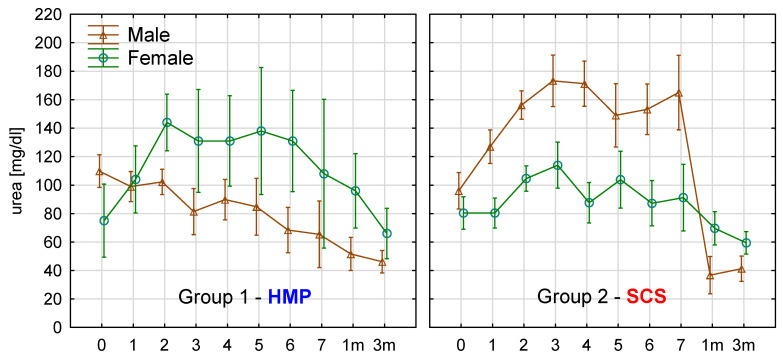
Urea concentration repeated measures with recipient gender and study groups (rANOVA, ***p* = 0.02138**, *n* = 66).

**Figure 10 antioxidants-10-01263-f010:**
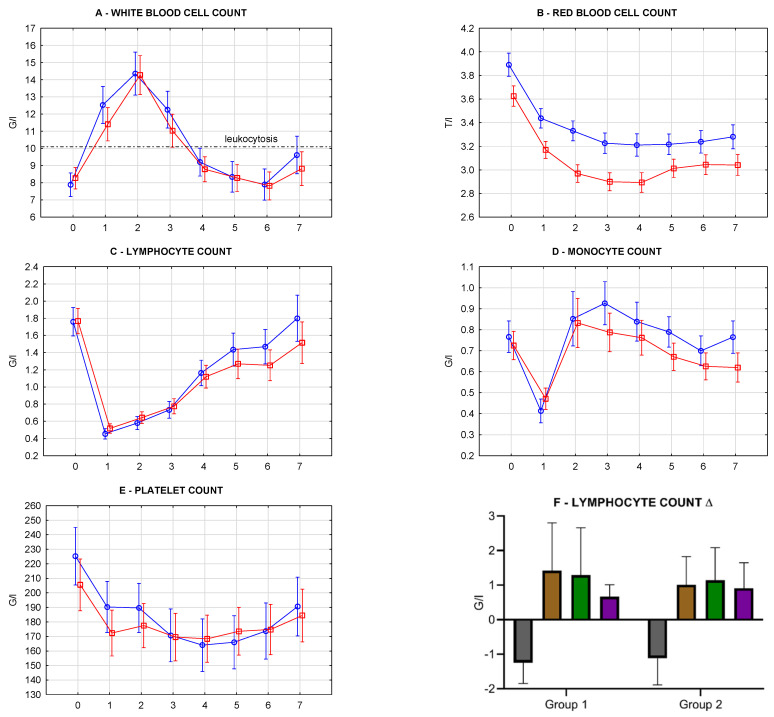
Mean blood count in following measures—7-day observation (rANOVA, *N* = 66, 
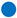
 Group 1 (HMP) *n*1 = 26, 

 Group 2 (SCS) *n*2 = 40, T—SE: (**A**). white blood cell count (WBC) (*p* = 0.70), (**B**). red blood cell count (RBC) (*p* = 0.881), (**C**). lymphocyte count (LYM) (*p* = 0.600), (**D**). monocyte count (MON) (*p* = 0.634), (**E**). platelet count (PLT) (*p* = 0.473).); (**F**). lymphocyte count relative changes (∆), (*t* test for independent groups, CI 95%): 
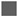
 day1–day0 (*p* = 0.461), 
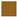
 day7–day1 (*p* = 0.185), 
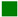
 month1–day1 (*p* = 0.652), 

 month3–day1 (*p* = 0.298).

**Table 1 antioxidants-10-01263-t001:** ELISA kits used for measurements of oxidative stress biomarkers.

Kit Name	Abbr.	Item No.	Manufacturer
Catalase Assay Kit	CAT AK	707002	Cayman Chemical
Glutathione Assay Kit	GSH AK	703002	Cayman Chemical
Glutathione Peroxidase Assay Kit	GPX AK	703102	Cayman Chemical
Glutathione Reductase Assay Kit	GR AK	703202	Cayman Chemical
Glutathione S-Transferase Assay Kit	GST AK	703302	Cayman Chemical
Malondialdehyde Assay Kit	MDA-586™	21044	OxisResearch™
Superoxide Dismutase Assay Kit	SOD AK	706002	Cayman Chemical
Thiobarbituric Acid Reactive Substances Assay Kit	TBARS AK	10009055	Cayman Chemical

**Table 2 antioxidants-10-01263-t002:** Study variables.

Category	Continuous	Qualitative
donor	age, BMI, creatinine, GFR, perfusate pH, CIT	gender (M/F), storage method (HMP/SCS), preservation solution (HTK/Cos/SPS), cause of death (Table 2), HLA (Table 3)
recipient	age, BMI, hemodialysis time, PRA, WIT2	gender (M/F), kidney (L/R), AHT, DM, HLA (Table 3), recipient kidney disease (Table 4), immunosuppression (Table 4)
biomarkers	GSH, GPX, CAT, SOD, GR, GS, TBARS, MDA	
kidney function	Ur, Cr, Na, K, D	DGF, AR, death, Tac, CsA, MMF, steroids, Simulect, ATG, CMV, BKV, SSI, UTI
blood count	WBC, LYM, MON, PLT, RBC	

Legend: GFR—glomerular filtration rate, HTK—Custodiol HTK, Cos—CoStorSol™, SPS—SPS-1^®^ (UW Solution), PRA—panel reactive antibody, WIT2—warm ischemic time 2nd (during vascular anastomoses in KTx), L/R—left/right, AHT—arterial hypertension, DM—diabetes mellitus, HLA—human leukocyte antigen.

**Table 3 antioxidants-10-01263-t003:** Donor-related parameters in the study group qualified for organ procurement.

Mean ± SD/n (%)	Total *N* = 66	Group 1 *n*1 = 26	Group 2 *n*2 = 40	*p*
Age [years]	52.2 ± 11.95	54.29 ± 11.30	50.80 ± 12.35	0.307
BMI [kg/m^2^]	26.34 ± 4.25	26.39 ± 4.63	26.30 ± 4.04	0.934
Gender (M/F)	35 (53.0%)	14 (53.8%)	21 (52.5%)	0.915
	31 (47.0%)	12 (46.2%)	19 (47.5%)	
creatinine [mg/dL]	0.89 ± 0.43	0.92 ± 0.42	0.88 ± 0.43	0.671
GFR [mL/min/1.73 m^2^]	89.50 ± 30.13	86.05 ± 29.40	91.84 ± 30.87	0.502
Preservation solution HTK/Cos/SPS	33 (50.0%)	14 (53.8%)	19 (47.5%)	0.895
	27 (40.9%)	11 (42.3%)	16 (40.0%)	0.268
	6 (9.1%)	1 (3.8%)	5 (12.5%)	0.233
cause of death				
BVM	39 (59.1%)	15 (57.7%)	24 (60.0%)	0.925
TBI	22 (33.3%)	9 (34.6%)	13 (32.5%)	0.899
A	3 (4.5%)	1 (3.8%)	2 (5.0%)	0.833
I	2 (3.0%)	1 (3.8%)	1 (2.5%)	0.763
BBT	0 (0.0%)	0 (0.0%)	0 (0.0%)	NA
Kidney (L/R)	32 (48.5%)	15 (42.3%)	17 (42.5%)	0.228
	34 (51.5%)	11 (57.7%)	23 (57.5%)	
perfusate pH	6.90 ± 0.25	7.11 ± 0.17	6.77 ± 0.19	**<0.001**

Legend: BVM—brain vascular malformation, TBI—traumatic brain injury, A—asphyxia, I—intoxication, BBT—benign brain tumor.

**Table 4 antioxidants-10-01263-t004:** Recipient-related parameters in the study group qualified for KTx.

Mean ± SD/n (%)	Whole *N* = 66	Group 1 *n*1 = 26	Group 2 *n*2 = 40	*p*
Age [years]	54.18 ± 12.68	57.27 ± 11.70	52.18 ± 13.02	0.111
BMI [kg/m^2^]	25.95 ± 4.25	25.83 ± 3.57	26.03 ± 4.69	0.869
Gender (M/F)	37 (56.1%)	19 (73.1%)	18 (55.0%)	**0.025**
	29 (43.9%)	7 (26.9%)	22 (45.0%)	
Hemodialysis time [months]	38.21 ± 25.19	37.48 ± 23.75	38.71 ± 26.50	0.865
PRA [%]	16.56 ± 27.27	10.24 ± 23.95	20.84 ± 28.89	0.171
CIT [hours]	17.89 ± 6.93	19.49 ± 7.68	16.76 ± 6.21	0.125
WIT2 [minutes]	24.89 ± 7.25	25.58 ± 6.60	24.41 ± 7.25	0.557
HLA mismatch A/B/DR/sum	1.09 ± 0.66	1.18 ± 0.59	1.03 ± 0.71	0.381
	1.21 ± 0.62	1.23 ± 0.61	1.20 ± 0.63	0.872
	0.49 ± 0.54	0.27 ± 0.46	0.63 ± 0.55	**0.011**
	2.79 ± 0.90	2.68 ± 0.78	2.86 ± 0.97	0.457
IGF/DGF/NGF	34 (51.51%)	16 (61.54%)	18 (45.00%)	0.312
	29 (43.94%)	10 (38.46%)	19 (47.5%)	0.220
	3 (4.54%)	0 (0%)	3 (7.5%)	0.115
AR (y/n)	3 (4.55%)	0 (0%)	3 (7.50%)	0.153
	63 (95.45%)	26 (100%)	37 (92.50%)	
Death (y/n)	2 (3.03%)	0 (0%)	2 (5.00%)	0.247
	64 (96.97%)	26 (100%)	38 (95.00%)	
AHT (y/n)	20 (30.30%)	7 (26.92%)	13 (32.50%)	0.630
	46 (69.70%)	19 (73.07%)	27 (67.50%)	
DM (y/n/PTDM)	9 (13.64%)	5 (19.23%)	4 (10.00%)	0.336
	52 (78.79%)	20 (76.92%)	32 (80.00%)	0.414
	5 (7.57%)	1 (3.85%)	4 (10.00%)	0.198
CMV/BKV/SSI/UTI	1 (1.52%)	0 (0%)	1 (2.50%)	0.387
	2 (3.03%)	1 (3.85%)	1 (2.50%)	1.000
	2 (3.03%)	1 (3.85%)	1 (2.50%)	1.000
	27 (40.91%)	11 (42.31%)	16 (40.00%)	0.413
Simulect/ATG/none	32 (48.48%)	13 (50.00%)	19 (47.50%)	0.264
	6 (9.09%)	1 (3.85%)	5 (12.50%)	0.231
	28 (42.42%)	12 (46.15%)	16 (40.00%)	0.861
Tac/CsA	62 (93.94%)	23 (88.47%)	39 (97.50%)	0.133
	4 (6.06%)	3 (11.53%)	1 (2.50%)	
MMF	61 (92.43%)	24 (92.31%)	37 (92.50%)	0.977
	5 (7.58%)	2 (7.69%)	3 (7.50%)	
Steroids	66 (100%)	26 (100%)	40 (100%)	1.000

Legend: M/F—male/female, HLA—human leukocyte antigens, IGF—immediate graft function, DGF—delayed graft function, NGF—no graft function, PTDM—post-transplant diabetes mellitus, ATG—anti-thymocyte globulin, Tac—tacrolimus, CsA—cyclosporine, MMF—mycophenolate mofetil, y/n—yes/no.

**Table 5 antioxidants-10-01263-t005:** Cause of chronic kidney disease of recipients qualified for KTx.

n (%)	Whole *N* = 66	Group 1 *n*1 = 26	Group 2 *n*2 = 40	*p*
GN	26 (39.39%)	10 (38.46%)	16 (40.00%)	0.402
AHT	7 (10.60%)	2 (7.69%)	5 (12.50%)	0.577
ADPKD	9 (13.64%)	4 (15.38%)	5 (12.50%)	1.000
DM	2 (3.03%)	2 (7.69%)	0 (0%)	0.248
Other causes	7 (10.60%)	2 (7.69%)	5 (12.50%)	0.577
Unknown cause	15 (22.73%)	6 (23.08%)	9 (22.50%)	0.713

Legend: GN—glomerulonephritis, ADPKD—autosomal dominant polycystic kidney disease.

**Table 6 antioxidants-10-01263-t006:** Heatmap of matrix correlations between oxidative stress biomarkers and kidney function parameters in repeated measurements (Ur—urea, Cr—creatinine, Na^+^—sodium, K^+^—potassium, D—diuresis) with aggregation intensity color scale.

	General Group (*N* = 66)					Group 1—HMP (*n* = 26)					Group 2—SCS (*n* = 40)				
	Ur	Cr	Na^+^	K^+^	D	Ur	Cr	Na^+^	K^+^	D	Ur	Cr	Na^+^	K^+^	D
CAT [U/mgHb]															
SOD [U/mgHb]															
TBARS [umol/L]															
MDA [umol/L]															
GST [U/mL]															
GST [U/mgHb]															
GSH [umol/gHb]															
GPX [U/L]															
GPX [U/mgHb]															
GR [U/mL]															
GR [U/gHb]															
**>5.0**	**4.0–5.0**		**3.0–3.5**		**2.0–2.5**	**1.0–1.5**		**0.5**		**1.0–1.5**	**2.0–2.5**		**3.0–3.5**	**4.0–5.0**	**>5.0**

**Table 7 antioxidants-10-01263-t007:** Heatmap of matrix correlations between oxidative stress biomarkers and blood count in repeated measurements (WBC—white blood cell count, LYM—lymphocytes count, MON—monocytes count, PLT—platelets count, RBC—red blood cell count) with aggregation intensity color scale.

	General Group (*N* = 66)					Group 1—HMP (*n* = 26)					Group 2—SCS (*n* = 40)				
	WBC	LYM	MON	PLT	RBC	WBC	LYM	MON	PLT	RBC	WBC	LYM	MON	PLT	RBC
CAT [U/mgHb]															
SOD [U/mgHb]															
TBARS [umol/L]															
MDA [umol/L]															
GST [U/mL]															
GST [U/mgHb]															
GSH [umol/gHb]															
GPX [U/L]															
GPX [U/mgHb]															
GR [U/mL]															
GR [U/gHb]															
**>5.0**	**4.0–5.0**		**3.0–3.5**		**2.0–2.5**	**1.0–1.5**		**0.5**	**1.0–1.5**		**2.0–2.5**		**3.0–3.5**	**4.0–5.0**	**>5.0**

## Data Availability

The data supporting reported results can be provided by the corresponding author upon a reasonable request, due to it involves ethical issues.

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
