# Peer review of "Assessment of Oxidative Stress Markers in Hypothermic Preservation of Transplanted Kidneys"

_antioxidants, 2021, doi:10.3390/antiox10081263_

Round 1

Reviewer 1 Report

This is a revised version of the Tejchman et al. paper in which they explore oxidative stress markers in preservation liquid at the end of the preservation period.

            I had a hard time reading this work. It is very long, very detailed, and it makes it difficult to extract the core messages and/or follow the rationale of the study.

            I list below several points which are important, however my main remarks is that the authors need to clarify and simplify. Using several paragraphs to describe a marker, which ultimately does not change significantly between groups, is prone to lose the reader. I understand that there is not word limitation in an open access paper, however the key to communicate the value of your work is in simplifying and condensing.  The usual scientific paper is 4000-5000 word long, this one is way beyond that point. The analyses are all over the place: between groups, within groups, then back to between groups, and when a strong correlation exist within a group, the fact that it is not present between groups, or goes in opposite directions, decreases the importance of the results… so why not focus exclusively on correlation between the groups ?

            Style is hard to follow as well. The paragraph from line 1060 to 1088, discussing the relation between oxidative stress markers and kidney function, arguably one of the most important aim of this study, is all over the place.

            There are interesting narratives to expose in this work. For instance: “more intense acidosis with lower TBARS, lower SOD and higher MDA.(…) In contrary, data dispersion was relatively high and MDA correlation gave the opposite information, thus conclusion that acidosis had the influence on biomarkers was not strongly supported.” This actually woud be a very interesting point to discuss. What does the literature say about MDA, TBARS and SOD, and has this kind of dichotomy been described ? What could it bring to our understanding of organ preservation mechanisms ?

            I thus believe that the authors must take the time to consider their entire dataset and decide which narrative to pursue, which hypotheses to investigate, and refrain from wanting to use all of their data.

            Other remarks:

  • Indicate identity of the group in the fig/table legends: by the middle of the paper, one doesn’t remember which is HMP and which is static.
  • Some parameters appear to be quantified in erythrocytes (expressed in mgHb), however other were quantified in the fluid (U/L or U/mL)? In that case, the dilution factor must be taken into account ? Legend must specifiy the nature of the sample measured, as the reader doesn’t know where the data comes from.
  • Discussion: lot of details on the oxidative stress markers, but very little discussion on why only two markers followed the hypothesis

Some style errors or mistakes are indicative of non native English speakers, hence I recommend a check once the final version is done.

Author Response

Dear Reviewer,

We would like to thank you for your valuable comments. We considered all of them, which resulted in the manuscript reorganization, clarification, and broader explanation of disputed issues. We addressed the reviewers’ (R) questions (Q) with the answers (A) in the order of their appearance.

R1Q01: I had a hard time reading this work. It is very long, very detailed, and it makes it difficult to extract the core messages and/or follow the rationale of the study.

             I list below several points which are important, however my main remarks is that the authors need to clarify and simplify. Using several paragraphs to describe a marker, which ultimately does not change significantly between groups, is prone to lose the reader. I understand that there is not word limitation in an open access paper, however the key to communicate the value of your work is in simplifying and condensing.  The usual scientific paper is 4000-5000 word long, this one is way beyond that point. The analyses are all over the place: between groups, within groups, then back to between groups, and when a strong correlation exist within a group, the fact that it is not present between groups, or goes in opposite directions, decreases the importance of the results… so why not focus exclusively on correlation between the groups ?

            Style is hard to follow as well. The paragraph from line 1060 to 1088, discussing the relation between oxidative stress markers and kidney function, arguably one of the most important aim of this study, is all over the place.

            There are interesting narratives to expose in this work. For instance: “more intense acidosis with lower TBARS, lower SOD and higher MDA.(…) In contrary, data dispersion was relatively high and MDA correlation gave the opposite information, thus conclusion that acidosis had the influence on biomarkers was not strongly supported.” This actually woud be a very interesting point to discuss. What does the literature say about MDA, TBARS and SOD, and has this kind of dichotomy been described ? What could it bring to our understanding of organ preservation mechanisms?

             I thus believe that the authors must take the time to consider their entire dataset and decide which narrative to pursue, which hypotheses to investigate, and refrain from wanting to use all of their data.

R1A01: The resubmitted manuscript was after the first major revision. The initial version was shorter; however, we had to extend many sections according to the reviewers’ comments. We agree that the present version is long and the amount of information regarding certain subjects is than average; however, we wanted to avoid insufficient information, which unfortunately was indicated in the previous revision. There are two extremes in study presentation: directly to the point without mentioning obvious things and without a broad discussion about side subjects, or - detailed results, analysis, and commentary about every step and variable of the study. We agree that both extremes are wrong, yet we could not keep the manuscript in a more brief version due to major issues. They regarded results selection and presentation, methodology as well as statistical analysis. Moreover, they had the potential to undermine the correctness of the entire study. After the first revision and the analysis of all comments, we felt pressure to extend the information about the study material, variables, recalculate statistics, and then prove that none of the possible important relations were omitted. Thus, numerous analyses with no statistical significances were mentioned in the results.

Major reviewers’ issues regarded results: 1. Preservation solutions and their further effects. Initially, it was reported that groups did not differ statistically, and there were no other correlations. Finally, we had to include additional statistical analysis with methodology, results, and commentary. 2. Donor parameters, which potentially could influence the oxidative stress markers and kidney function. It was followed by introducing new tables presenting all variables and new statistical analyses of variables' cross-relations. New calculations were followed by selecting additional tests, considering combinations of continuous and qualitative variables, which we were induced to describe. 3. Relationship between sex and oxidative stress as well as kidney function and blood. We agree that there is an established relationship between sex and kidney function, so such an analysis in our study with additional oxidative stress assessment was an interesting issue; however, initially, we did not expect strong correlations in our relatively small examined group. Nevertheless, finally, we had to find the space for this, perform appropriate analyses, generate the figure and extend the discussion.

Reviewers also pointed out issues regarding methodology - obtaining and preparing the perfusate, source of erythrocytes, and dilution effect. We agree that the initial description was too brief and lacking sufficient information, thus after revision, those paragraphs had to be vastly extended. It took some space to describe how the material was collected, what fractions were separated and what exactly was the source of measurement. Nevertheless, even after broader correction, we did not avoid doubts, e.g., quantification expressed in mgHb or U/L. Moreover, we wanted to precede other potential underreported issues and meet other possible expectations by extending analogous information in the manuscript. We also had the experience of questioning certain major issues, which preventively resulted in, e.g., the detailed description of preservation fluids or chemical methods used in the assays.

Finally statistics. Reviewers questioned the number of statistical tests used. There were doubts about the usage of ANOVA and the nonparametric alternatives, distinctions between continuous and qualitative variables, methodology of heat maps, additional presentation methods - correlation plots, residuals, Q-Q plots, etc. Initially, a brief description turned into a very detailed one, justifying the choice, correctness, and the purpose of the used methods. We agree that the statistics subsection provide much information, more than we are used to, but statistical issues in the initial revision were quite serious in our feeling. We wanted to assure the readers that our statistical analysis was appropriate and far more complex than just an autopilot run on the statistical software with default settings.

We would like to express that the volume of the present manuscript was dictated in a significant part by the reviewers’ requests and suggestions. We had to address all critical issues after recommendations, including, e.g., “need to be included in the statistical analysis”, “it needs to be investigated and discussed”, “the correlation between […] is difficult to support“, “The authors must present […] in order for the reader to fully estimate the value of this analysis.”, “This must be investigated and demonstrated with the proper tools.”, “need to be determined by hazard ratio and heatmap” “the statement […] need to be clarified or corrected”. We are aware that experts seek different information in scientific articles than less experienced “explorers”. Obviously, briefness is crucial for an expert, but depending on the reader, it seems that it differs a lot where is the borderline between information noise and insufficient information. We believe that achieving the balance between briefness and the total exhaustion of the topic is quite hard, although we did our best to correct the manuscript with the reviewer’s suggestions in mind. We admit that the final version gravitates a bit more towards broader descriptions, but after all reviewers’ comments analysis we agreed, that confusion caused by insufficient information was greater than caused by the excessive descriptions; thus, we were cautious in limiting them.

We revised language, corrected spelling, and reorganized discussion, where oxidative stress became a separate subsection. In addition, unnecessary parts of the manuscript were moved or removed, including methods, statistics and discussion. Some additional information was given to clarify particular subjects.

R1Q02: Indicate identity of the group in the fig/table legends: by the middle of the paper, one doesn’t remember which is HMP and which is static.

R1A02: It was corrected in the selected Figures and the manuscript, including discussion and conclusions.

R1Q03: Some parameters appear to be quantified in erythrocytes (expressed in mgHb), however other were quantified in the fluid (U/L or U/mL)? In that case, the dilution factor must be taken into account ? Legend must specifiy the nature of the sample measured, as the reader doesn’t know where the data comes from.

R1A03: All oxidative stress markers were measured in the cellular material centrifuged after perfusate collection, thus in erythrocytes. The cellular part, after centrifuging and preparation, became a homogenous solution containing hemoglobin. Enzyme activities were calculated per volume of the above solution or hemoglobin mass in the same solution. Those volumes had nothing to do with the fluid part of the perfusate, which was separated after centrifuging and was used only for pH measurement. The dilution effect influenced the pH, but it was irrelevant to oxidative stress markers. Non-enzymatic biomarkers were calculated in the same solution as enzymatic ones. The additional sentence was given in the methods section.

R1Q04: Discussion: lot of details on the oxidative stress markers, but very little discussion on why only two markers followed the hypothesis

R1A04: The answer was given mainly in the limitations subsection of the discussion; however, it was pretty brief. We chose to examine the ischemic part of IRI, where material for measurements of oxidative stress biomarkers was obtained at the end of kidney storage. It is quite a unique approach, however noninvasive, which was crucial in our study design. In reference studies regarding kidney transplantation and oxidative stress, we are usually read about the evaluation of full IRI, emphasizing the reperfusion phase. Reoxygenation is a critical phenomenon in IRI and oxidative stress; however, we evaluate the period before surgical procedures in our study. We believe this is a critical factor in our analysis. Additional explanation was given in the discussion.

R1Q05: Some style errors or mistakes are indicative of non native English speakers, hence I recommend a check once the final version is done.

R1A05: The manuscript was revised in that matter.

With best regards

Katarzyna Kotfis

Reviewer 2 Report

Authors provided satisfactory responses to my concerns and the manuscript is upgraded according to suggestions.

Author Response

Dear Reviewer,

Thank you for your kind evaluation of our revised manuscript.

As suggested, we performed a final spell check.

With best regards

Katarzyna Kotfis 

Reviewer 3 Report

Authors provide a very comprehensive study about possible benefits of hypothermic perfusion in solid organ preservation to prevent/reduce ischemia-reperfusion injury. Although this problem was discussed recently (Giraud S, Thuillier R, Cau J, Hauet T. In Vitro/Ex Vivo Models for the Study of Ischemia Reperfusion Injury during Kidney Perfusion. Int J Mol Sci. 2020 Oct 31;21(21):8156. doi: 10.3390/ijms21218156), Authors study improves our knowledge on oxidative stress changes related with different methods of organ storage. It also should be noted that we do not have many studies on humans, most data come from animal studies.

However please try to correct the manuscript, it looks like each part was written by different person, so unintentional errors occur:

1) Please decide what is the final title of the manuscript, the one chosen in the submission center is "The role of oxidative stress markers in hypothermic preservation of transplanted kidneys ",

2) Please if possible rearrange the abstract, trying to show the results by using so many arrows up or down is not helping to understand your results; also please decide whether you analyzed 'concentrations of enzymes' or their activity (lines 19-20), concentration does not have any meaning in enzymatic analysis,

3) please think about changing the sentence in lines 44-45, other forms of kidney replacement therapy, like hemodialysis and peritoneal dialysis can also be considered as 'curative', however every kidney replacement therapy, even kidney transplantation is trying to restore kidney function with different effect, so I think its hard to say that KTX is 'the only curative treatment option' for patients with ESKD,

4) please check the abrreviation AR in the manuscript and correct it, it should be related to acute rejection (line 509), not autoimmunological rejection (line 55),

5) please correct the way of writing chemical names of agents, like calcium chloride 2 H2O (line 125),

6) please explain abbreviations when used for the first time, like HEPES (4-(2-hydroxyethyl)-1-piperazineethanesulfonic acid) or explain what is it, similar CMV, BKV, UTI, SSI (line 268) occur faster than their explanation in the Table 2, similar is with AHT and DM in Table 4,

7) please correct the abbreviation HES, pentafraction is not the hydroxyethyl starch, but its fraction,

8) please try to shorten the manuscript, in the laboratory analysis section the technical desciption of each method is too long and not necessary,

9) in the laboratory analysis section you should describe analytical details, not describing each enzyme and its role,

10) the statistical analysis should contain shorter information about used tests, your hypothesis/questions should be moved to introduction or discussion and written and simplier way,

11) traumatic brain injury should be abbreviated as TBI, not TMI (line 502),

12) some data, about well known things, like WBC consists of NEU+LYM+EOS+MON+BAS are really not necessary, especially in the discussion (lines 1116-1118),

13) in the conclusion section I would like to have more clear data, not information about group 1 and group 2, you should be more specific and write about perfusion method, not 'group".

Author Response

Dear Reviewer,

We would like to thank you for your valuable comments. We considered all of them, which resulted in the manuscript reorganization, clarification, and broader explanation of disputed issues. We addressed the reviewers’ (R) questions (Q) with the answers (A) in the order in order of their appearance.

R3Q01: Please decide what is the final title of the manuscript, the one chosen in the submission center is "The role of oxidative stress markers in hypothermic preservation of transplanted kidneys”

R3A01: The manuscript was initially submitted under the title “The role of oxidative stress markers in hypothermic preservation of transplanted kidneys”. There was an issue regarding the word “role”, so the title was changed to “Assessment of oxidative stress markers in hypothermic preservation of transplanted kidneys”. Probably submission exists in the system under the initial title. The correct one is in the revised manuscript.

R3Q02: Please if possible rearrange the abstract, trying to show the results by using so many arrows up or down is not helping to understand your results; also please decide whether you analyzed 'concentrations of enzymes' or their activity (lines 19-20), concentration does not have any meaning in enzymatic analysis.

R3A02: The arrows were corrected in the first revision in all sections except the abstract. Nevertheless, we agree that they remained confising also in the abstract. Arrows were corrected. Word “concentration” regarding GPX and enzymes was a mistake. It should be an “activity”. It was corrected. “Concentration” regarded only MDA, TBARS, and GSH.

R3Q03: Please think about changing the sentence in lines 44-45, other forms of kidney replacement therapy, like hemodialysis and peritoneal dialysis can also be considered as 'curative', however every kidney replacement therapy, even kidney transplantation is trying to restore kidney function with different effect, so I think its hard to say that KTX is 'the only curative treatment option' for patients with ESKD.

R3A03: We agree that the word “curative” was inappropriately used. We were thinking more like a “causative”; however, the sentence was entirely changed.

R3Q04: Please check the abrreviation AR in the manuscript and correct it, it should be related to acute rejection (line 509), not autoimmunological rejection (line 55)

R3A04: “Autoimmunological” was a mistake. It was corrected to “acute”.

R3Q05: Please correct the way of writing chemical names of agents, like calcium chloride 2 H2O (line 125).

R3A05: They were corrected to full chemical names.

R3Q06: Please explain abbreviations when used for the first time, like HEPES (4-(2-hydroxyethyl)-1-piperazineethanesulfonic acid) or explain what is it, similar CMV, BKV, UTI, SSI (line 268) occur faster than their explanation in Table 2, similar is with AHT and DM in Table 4,

R3A06: HEPES was 4-(2-hydroxyethyl)-1-piperazineethanesulfonic acid. It was corrected in the manuscript. CMV, BKV, UTI, SSI were explained earlier in the manuscript and descriptions under Table 2 were removed. AHT and DM first time show up in Table 2 in description of qualitative recipient factors and they are explained in the legend. Descriptions were removed from Table 5.

R3Q07: Please correct the abbreviation HES, pentafraction is not the hydroxyethyl starch, but its fraction.

R3A07: It was corrected.

R3Q08: Please try to shorten the manuscript, in the laboratory analysis section the technical desciption of each method is too long and not necessary.

R3A08: It was corrected. Technical descriptions were limited or removed.

R3Q09: In the laboratory analysis section you should describe analytical details, not describing each enzyme and its role.

R3A09: It was corrected. Descriptions were removed.

R3Q10: The statistical analysis should contain shorter information about used tests, your hypothesis/questions should be moved to introduction or discussion and written and simpler way.

R3A10: It was corrected. Some descriptive parts were removed. Questions were moved to introduction and simplified.

R3Q11: Traumatic brain injury should be abbreviated as TBI, not TMI (line 502).

R3A11: It was corrected.

R3Q12: Some data, about well known things, like WBC consists of NEU+LYM+EOS+MON+BAS are really not necessary, especially in the discussion (lines 1116-1118).

R3A12: We thought so in the initial manuscript; however, there was an issue regarding the role of neutrophils in leukocytosis, after which we decided to put a full description. We have left the explanation and removed the basics.

R3Q13: in the conclusion section I would like to have more clear data, not information about group 1 and group 2, you should be more specific and write about perfusion method, not 'group".

R3A13: It was corrected.

With best regards

Katarzyna Kotfis MD, PhD, DESA

Round 2

Reviewer 1 Report

This is the second revised version of the Tejchman et al. paper in which they explore oxidative stress markers in preservation liquid at the end of the preservation period.

I read with interest the answer to the reviewer, and I must disagree with the authors regarding the reasons for keeping a lengthy and meandering text. On must be able to synthetize the results into a coherent demonstration. If indeed this reviewers required the authors to check for confounding variables at the donor level and elsewhere, the authors can perform such verifications and briefly mention their results in the text, while providing the more complete analysis in a supplementary data file.

As it is, the reader must follow a tortuous path to end with a negative conclusion:

“There were statistically significant correlations between oxidative stress markers and kidney function; however, they could not confirm that kidney function after transplantation is related to oxidative stress markers measured in preservation time. There were certain alterations of blood count observed in repeated measures after KTx; however, they seemed to relate to other factors than oxidative stress or acidosis. Despite few statistically significant relations, the influential role of recipient gender, gender matching, preservation solution, and perfusate pH was not confirmed.”

Which is pretty frustrating. I agree that negative data is still data, however it must be clearly displayed.

I suggest to the reviewer to consider their hypotheses:

“We attempted to support the study hypothesis that the method of kidney storage plays a role in kidney transplantation outcome”

Considering this was already done, on much wider cohorts, and showed that HMP as mostly helpful, and only in a limited scope, on fragile organs such as ECD, is their experimental set up appropriate ?

Only urea was different between the groups, a marker that is not acknowledged as a strong postransplantation endpoint, but it is still something to work with.

“We also tried to support the hypothesis that the beneficial effect of HMP on transplanted kidneys may rely on limiting the negative impact of IRI and oxidative stress in the period preceding reperfusion. “

Of all the oxidative stress markers investigated, only two were statistically different between the group. Three if we include pH. Why then consider all the other markers in further analysis ?

In my opinion, the fact that the authors explored the influence of all the variables within each group does not make sense regarding the hypotheses of their work.

For instance: “There was a statistically significant correlation between higher pH and lower donor’s BMI in Group 1 (R Spearman, p<0.05) as well as higher donor’s age in Group 2 (correlation matrix – Pearson test, p=0.045) (Figure 3). “

How is that relevant to the hypotheses which the authors decided to explore, as exposed in the introduction ?

It is stated that the authors want to confirm HMP is more protective that SCS and leads to better outcome, the authors confirm that with urea. This reviewer wanted to check if donor factors were not confounding this confirmation, hence the authors must indeed show that there is not statistical different in BMI, age, etc between the groups. The table shows there isn’t, which is sufficient, but if you want to take it further one might consider testing the correlation between BMI and urea (as a whole, not within a group). But no more. And the later correlation can be relegated to supplementary data.

The authors have thus one functional parameter: urea. Three parameters differ between HMP and SCS: pH, MDA, GPX. Are they correlated with outcome ? Not within group, but overall. If it is the case, then your second hypothesis would be confirmed.

Reviewer 3 Report

Thank you very much, Authors improved significantly the manuscript and answered to my questions.

Author Response

Thank you for a positive comment.

We tried to do our best to improve the manuscript.

With best regards

Katarzyna Kotfis 

This manuscript is a resubmission of an earlier submission. The following is a list of the peer review reports and author responses from that submission.

Round 1

Reviewer 1 Report

The work presented by Tejchma K et al, entitled: The role of oxidative stress markers in hypothermic preservation of transplanted kidneys, is an extensive descriptive statistical study in which two indicators of lipid peroxidation oxidative injury (TBARS and MDA), antioxidant enzymes and glutathione were measured in preservation fluid in kidney transplantation. In addition, hematological and renal function parameters were measured between 1 day and 3 months after transplantation.

The main observations are:

1- The introduction is very long. Lines 76-112 include a list of measured variables that could be in the methods.

2- Figure 1 is complex. It includes too much information, variables, and arrows that are confusing.

3- Discussion should try to explain the results obtained, especially with regard to oxidative stress.

4- It would be better to include an oxidative stress indicator of lipid peroxidation and another of protein damage, e.g., carbonilation.

Reviewer 2 Report

In this paper, Tejchman et al. explore oxidative stress markers in preservation liquid at the end of the preservation period.

First I’d like to congratulate the authors on their initiative and work, because logistically this work must have been quite difficult to perform, especially regarding sample collection. However, there are several issues with the data and the conclusions drawn from it which lead me to request major revision before it can be considered for publication.

Major issues:

  • It is not clear which solution was used in the machine. Indeed, the lifeport is designed to run with KPS, not custodial or UW. Did the authors use the later on the Lifeport ? What is the rationale ?
  • As stated in the introduction, machine perfusion is particularly effective when used in ECD or DCD donor organ. Why thus did the authors perform their study on SCD donor organs ?
  • Regarding the donors, no evaluation of donor parameter on the results is presented. Many of these parameters can actually influence oxidative stress (cause of death, length of reanimation, therapeutics, kidney function, etc) and need to be included in the statistical analysis. One parameter mentioned in the discussion is age, and it relationship with pH. If indeed there is a relation, it needs to be investigated and discussed, as it could bias the results.
  • The authors measure in the perfusate markers and enzymes which are normally present intracellularly. What is the source of these markers in the perfusate ? This needs to be discussed.
  • The correlation between pH and TBARS is difficult to support, as there is a lot of spreads in the data points. The authors must present residual plots and q-q plots in order for the reader to fully estimate the value of this analysis.
  • Regarding data presentation, use of dot plot instead of histograms is essential to better appreciate the dispersion of the data, particularly considering the limited differences observed.
  • The terms trend and/or tendency is abusively used when the data only reflects equivalent measurements between the groups. The authors must re-word their conclusions
  • Why so many test to establish differences between two groups? is there a rationale for using all three ? The one used to determine significance must be stated in the figure. Moreover, n must also be stated in the legend.
  • Significance of variations over time are estimated by c2 ANOVA, whereas Friedmann is mentioned in the methods. Which is it?
  • 2 way anova is used to compare continuous data (such as urea, creatinine), whereas this is not the proper use for this test as one of the variable is continuous (time). Favor the use of area under the curve comparison for these.
  • Why use weighted means to describe functional data ? Where are the standard deviations ?
  • Correlation matrices are a nice ideas, but they are very confusing. It’s difficult to extract useful information from this data, as each group are separated. If for instance the authors could show that within a group, a parameter measured at time of transplantation (for instance TBARS) is correlated with serum creatinine level at day 7, or better yet with area under the curve serum creatine, or with speed of decrease, it would represent an interesting area of investigation. This must be investigated and demonstrated with the proper tools (such as correlation plots, residuals, etc).
  • There are too many variables investigated in the correlation matrices, most notably the functional values at each time point. As stated above, when using a continuous/repeated measure variable, there is no independence between time points, they thus cannot be considered separately. Use these parameters in conformations which can represent is evolution over time: area under the curve, speed of decrease between time points, delta between time points, etc.
  • The questions behind the correlation matrices are also unclear. Is the aim to evaluate the organ at transplantation and anticipate its outcome ? If so, it needs to be investigated and discussed.
  • The dilution effect is barely discussed, whereas is appears to be a major bias in the analysis: during static storage, all the makers are concentrated in the liquid remaining in the kidney, whereas during dynamic storage these are diluted in the full volume of the solution. Did the authors consider normalization strategies ?
  • The solution effect is not investigated, although there a major differences between Custodiol and UW. This needs to be included in the analysis (in this setting the 2 way ANOVA is justified)
  • The source of erythrocytes is confusing. Where there a lot of blood cells remaining ? Are there differences between machine perfusion and static in cell quantity alone ?
  • The use of arrow in the abstract and the text is confusing, for instance (¯ urea and ­ MDA, ¯ CAT, ¯ TBARS). What is the aim here ? Especially since this is within a group, and until then the paper was aiming at differentiating between these two groups, not exploring the dispersion within each… This is confusing and frustrating for the reader, the authors must better explain their rationale
  • the work does not actually fit its title: the authors do not investigate the role of the measured markers. Consider choosing a more appropriate title.

Minor issues

  • In the methods, the authors mentioned that samples were thawed at room temperature. This is not recommended as variation in thawing time can influence the integrity of the sample. Have the authors performed quality analysis on their sample?

            In all, there is a lot of promise in this paper but the methodology, the way the data is displayed, and the conclusions drawn need to be revised in depth in order for this work to be considered for publication.

Ps: several typos:

-          2.3. Post-transplant care : L.242 moNths instead of moths

-          2.4. Laboratory analysis : L.325 TBARS (S is missing)

-          3.1 Examined groups : Table 3, the definitions of HA and ATG are missing in the legend

-          3.2 The pH of the perfusate : Figure 2A : please keep the same color for group 1 (black) and 2 (grey) throughout the article to make the article easier to read

-          4.2 L. 633 S is missing at the end of seem (mais je ne suis pas trop sûre…)

-          4.2 L 645 the word “in” has to be removed

Reviewer 3 Report

Tejchman et al. investigated two groups of renal transplant recipients in relation to the method of kidney storage using hypothermic machine perfusion (HMP) and static cold storage (SCS) and analysed the correlation of oxidative stress markers of perfusate. Authors have found that early blood urea is significantly increased in SCS recipients compared to HMP recipients. Several prior studies suggested that female recipients may have a graft survival advantage over males (Lepeytre et al. J Am Soc Nephrol. 2017;28(10):3014-3023). There are significant large female patients (~76%) have received SCS kidney (table-3)  in this manuscript and data distribution is not clear in graphs. Whether the detrimental effects on SCS recipients are due to SCS or sex is not clear and need to be clarified in this manuscript.

Specific comments:

  1. Relationships in between oxidative stress markers and sex is need to be determined by hazard ratio and heatmap.
  2. Relationships in between kidney functions, blood counts and sex are need to be determined by hazard ratio and heatmap.
  3. Blood count and blood morphology are two distinct terms in haematology. Authors have analysed the recipients blood count not morphology in the entire manuscript and need to be corrected.
  4. Correction of spelling/ typo of hypothermic is required in line 63, page 2.
  5. Line 744- authors have stated that ---most of recipients in both groups developed leukocytosis caused by neutrophils (Figure 7, A)- the statement and figure 7A are need to be clarified or corrected.
  6. Concise description discussion will attract more readers and enhance the quality of the manuscript.